# Position: Stop Chasing the C-index
# when Evaluating Survival Analysis Models

**Christian Marius Lillelund** [1 2]  **Shi-ang Qi** [1 3 4]  **Russell Greiner** [1 3]  **Christian Fischer Pedersen** [2]

## Abstract

The current state of evaluation in survival analysis is plagued by the persistent use of evaluation metrics in ways that are misaligned with the stated modeling objective. In addition, many such evaluations are based on censoring assumptions that are left implicit or unjustified. This means that the reported performance can be misleading and may fail to answer the scientific or modeling question the evaluation was intended to address. In this position paper, we critically examine evaluation practices in survival analysis and highlight how censoring makes evaluation fundamentally different from standard regression or classification. We place particular focus on concordance-based measures, such as the C-index, which we show are heavily overused in the literature. To help identify appropriate metrics, we propose a set of key desiderata and introduce a double-helix ladder, in which valid evaluation requires alignment between metric and modeling assumptions. Through controlled experiments, we show that violations of this alignment can lead to misleading model comparisons. We conclude by providing practical guidance on how to evaluate a survival model.

## 1. Introduction

Survival analysis – which studies ways to predict when an event will occur – is widely used in domains such as healthcare (Cohen et al., 2008; Lee et al., 2021), mechanical engineering (Zhang et al., 2019), and economics (Ahn, 2023). A defining characteristic of survival data is *censoring*, where the event time is only partially observed for some instances.

The most common case is *right-censoring*, in which only a *lower bound* of the event time is known – *e.g.*, when a patient has not yet experienced disease progression by the end of a clinical study. Because censored observations appear in the test set (as well as the training set), performance metrics must match the prediction objective and account for censoring to be meaningful. In practice, however, these considerations are often overlooked:

**First, metrics are often misaligned with the stated objective**: many projects use measures that use naive discrimination by default, even when the real objective of the project is different – *e.g.*, accurate time-to-event or probabilistic prediction. **Second, these metric choices are often made without considering of censoring or correcting for it.** Many projects proceed as if censoring were "random" – that is, independent of the event process – an assumption we define precisely below, yet such assumptions are rarely stated, justified, or adjusted for empirically. Because censored observations appear in both training and test sets, evaluation metrics must explicitly account for censoring to avoid biased performance estimates. In this setting, censoring is not a technical nuisance but a structural feature that determines whether an evaluation metric is meaningful.

Zhou et al. (2023) found that over 80% of studies in survival analysis used the *C-index* as the primary metric. The C-index (CI) is a discrimination metric: it measures whether predicted risk scores correctly rank individuals by event time[1]. While the limitations of the C-index are well documented (Hartman et al., 2023), our findings suggest that its widespread use points to a broader issue: a review of 92 methodological and application-based survival papers, from 2023 to 2025 (Appendix A), reveals that **approximately 72%** rely on evaluation metrics that do not align with their stated objectives, and often do so without stating their censoring assumptions (*e.g.*, random) or attempting to correct

---

[1]Department of Computing Science, University of Alberta, Edmonton, Canada [2]Department of Electrical and Computer Engineering, Aarhus University, Aarhus, Denmark [3]Alberta Machine Intelligence Institute, Edmonton, Canada [4]Vector Institute, Toronto, Canada. Correspondence to: Christian Marius Lillelund <clillelund@ualberta.ca>.

*Proceedings of the 43rd International Conference on Machine Learning*, Seoul, South Korea. PMLR 306, 2026. Copyright 2026 by the author(s).

---

[1]At a high level, each instance is represented by a feature vector $\boldsymbol{x}_i \in \mathcal{X}$, where typically $\mathcal{X} = \mathbb{R}^d$. A model $M(\cdot)$ predicts a scalar risk score $M(\boldsymbol{x}_i) \in \mathbb{R}$. For two instances $\boldsymbol{x}_i$ and $\boldsymbol{x}_j$, we interpret $M(\boldsymbol{x}_i) > M(\boldsymbol{x}_j)$ as predicting that instance $i$ will experience the event (*e.g.*, die) earlier than instance $j$. The C-index then measures how often this ranking is correct among comparable pairs, *i.e.*, pairs where one observed event precedes the other time. See Appendix B for a formal definition.

for censoring to make those same metrics valid. Figures 1 and 2 summarize these findings.

**Position.** We argue that a substantial portion of recent survival analysis research is evaluated incorrectly, leading to misleading model comparisons and overstated performance claims. This arises from a systematic mismatch between modeling objectives, evaluation metrics, and the censoring assumptions those metrics rely on. In practice, discrimination-based metrics (in particular the C-index) are often misused to support claims about time-to-event prediction or probabilistic calibration, while censoring is assumed to be random (Definition 2.1) with little justification. To make this mismatch explicit, we introduce a ladder hypothesis between models, metrics, and censoring assumptions. Through controlled experiments, we show that popular metrics become increasingly misleading as censoring deviates from the common assumption of randomness. While prior work has discussed why and when the C-index can be problematic, our goal is not only to critique existing practice but to offer practical guidance for meaningful evaluation. In particular, we argue that evaluation should start from the research objective, make explicit the censoring assumptions one is willing to defend, and ideally select metrics that are valid under those assumptions.

| Section | Content |
|---------|---------|
| Sec. 2 | Survival analysis and censoring assumptions. |
| Sec. 3 | Examples of model-metric mismatch and five desiderata for evaluating survival models. |
| Sec. 4 | Aspects of evaluation and metrics. |
| Sec. 5 | The Ladder Hypothesis and practical experiments illustrating model-metric mismatch. |
| Sec. 6 | Recommendations for aligning evaluation choices with study objectives and censoring assumptions. |
| Sec. 7 | Alternative views on model evaluation. |

Code to run the experiments is available at: https://github.com/thecml/position-cindex.

## 2. Survival Analysis

### 2.1. Survival Data

We consider a survival dataset $\mathcal{D} = \{(\boldsymbol{x}_i, t_i, \delta_i)\}_{i=1}^{N}$, where $\boldsymbol{x}_i \in \mathbb{R}^d$ denotes covariates, $t_i \in \mathbb{R}_+$ the observed time, and $\delta_i \in \{0, 1\}$ the event indicator. At the population level, let $\boldsymbol{X} \in \mathcal{X} \subseteq \mathbb{R}^d$ denote the covariate vector, and let $E$ and $C$ denote nonnegative event-time and censoring-time random variables, with realizations $e_i, c_i$. We consider right-censored data, where

$$T := \min\{E, C\}, \qquad \Delta := \mathbb{1}[E \leq C].$$

Thus, each observed triplet $(\boldsymbol{x}_i, t_i, \delta_i)$ realizes $(\boldsymbol{X}, T, \Delta)$. We assume these triplets are drawn i.i.d. from $\Pr(\boldsymbol{X}, T, \Delta)$, equivalently induced by $\Pr(\boldsymbol{X}, E, C)$. Here, we focus on

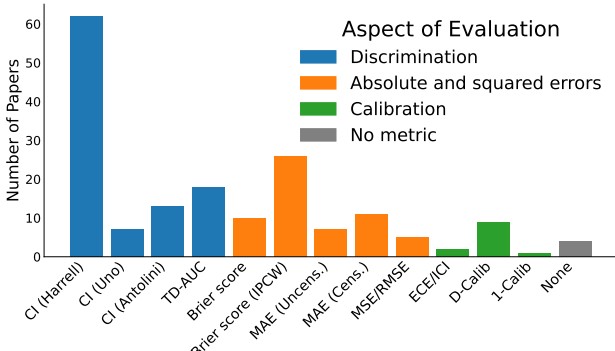

*Figure 1.* Overview of metric usage in survival analysis publications (2023-2025, see Appendix A). Papers reporting multiple metrics contribute one count to each metric. Note that most papers rely on metrics that only assess a model's discrimination (*i.e.*, ranking) ability, not how well it can make accurate time-to-event estimates nor provide calibrated survival probabilities.

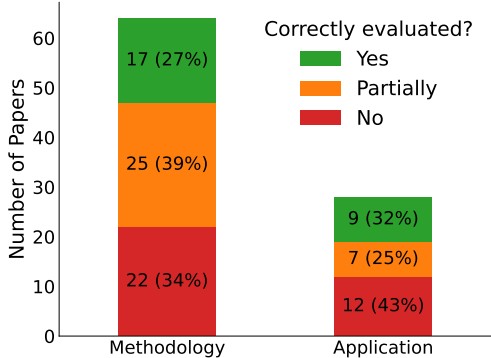

*Figure 2.* Overview of metric correctness in survival analysis publications (2023-2025; see Appendix A). Each paper is labeled **Yes**, **Partially**, or **No** depending on whether the reported evaluation metrics align with the stated modeling objective and whether censoring assumptions are stated, justified or corrected for to make those metrics valid. We find that 73% of surveyed methodology papers and 68% of application-focused papers do not fully satisfy these criteria.

survival models that estimate an individual survival distribution (ISD), an instance-specific survival function describing the probability that the event has not occurred beyond each time point (Haider et al., 2020). Given $\boldsymbol{x}_i$, an ISD model $M(\cdot)$ produces $M(\boldsymbol{x}_i) = \widehat{S}(\cdot \mid \boldsymbol{x}_i)$, where

$$\widehat{S}(t \mid \boldsymbol{x}_i) \approx \Pr(E > t \mid \boldsymbol{X} = \boldsymbol{x}_i), \qquad t \geq 0.$$

ISDs support time-to-event estimates, pointwise event probabilities, and derived risk scores. We refer to these outputs as prediction targets.

### 2.2. Censoring Assumptions

We typically distinguish three assumptions about the relationship between $E$ and $C$ (see Figure 3), which is generally not identifiable from the observed data (Tsiatis, 1975).

These assumptions affect both the validity of a survival learner[2] and its evaluation metric.

**Definition 2.1** (**Random Censoring**). Instances censored at time $t$ are representative of all instances still at risk at $t$, in terms of their survival experience (Kleinbaum & Klein, 2012, Ch. 1). For example, in a medical study, patients are censored because the study ends, or because they move away for reasons unrelated to health. These are assumed to have the same subsequent death rate as uncensored patients who remain in the risk set[3]. Formally, $E \perp\!\!\!\perp C$[4].

**Definition 2.2** (**Independent Censoring**). Within any sub-group of interest (*e.g.*, smokers or non-smokers), instances censored at time $t$ are representative of those who remain at risk at $t$ with respect to their survival experience (Kleinbaum & Klein, 2012). This is also called conditional independent censoring, since observed covariates are assumed to explain the event–censoring relationship. In a cancer study, tumor grade may affect both survival and loss to follow-up, so censoring may be associated with event time marginally. However, if loss to follow-up is unrelated to survival within each tumor-grade group, censoring is independent conditional on tumor grade. Formally, $E \perp\!\!\!\perp C \mid \boldsymbol{X}$[5].

**Definition 2.3** (**Dependent Censoring**). Within any sub-group of interest, instances censored at time $t$ are *not* representative of those who remain at risk at time $t$ with respect to their survival experience. Equivalently, censoring is dependent when it remains related to the event process even after conditioning on the observed covariates. In the afore-mentioned cancer study, tumor grade may be recorded, but unobserved frailty, socioeconomic status, or treatment tolerance may affect both survival and dropout from follow-up. Then, even within each tumor-grade group, censored patients may have different survival prospects from those who remain under observation. Formally, $E \not\perp\!\!\!\perp C \mid \boldsymbol{X}$.

## 3. Why Evaluation Matters

Censoring is not a single mechanism, but may behave in various ways between datasets. After a learner has learned from a dataset that includes censored instances, the resulting model will then be evaluated against held-out data, which

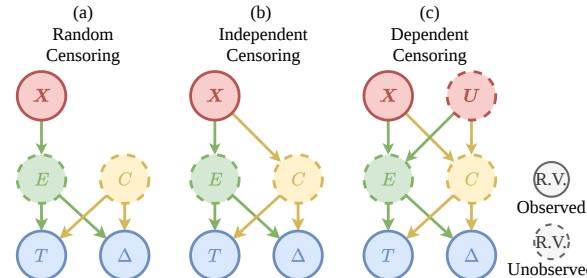

*Figure 3.* Comparison of three censoring assumptions, illustrated using directed acyclic graphs (DAGs) – where directed arcs show statistical dependence (Pearl, 2009). (a) Random censoring involves a random censoring time unrelated to the event time. (b) Independent censoring assumes that the censoring time is statistically independent of the event time given the observed covariates. (c) Dependent censoring violates this assumption as an unobserved confounding variable induces dependence between $E$ and $C$.

also includes censored instances[6]. The key distinction is whether censored individuals remain representative of those still under observation marginally, only after conditioning on observed covariates, or not even after such conditioning.

### 3.1. Objective-Metric Mismatch

Our analysis shows that the community does not always carefully consider how their models are evaluated or which criteria are used to assess them. Here are three examples.

Steinberg et al. (2024) presented a foundation model called *MOTOR*, designed to predict time-to-event outcomes from 55M patient records. The authors focused on accurately predicting the time to a specific event, such as when a particular diagnosis code will be assigned to a patient, but did not directly evaluate time-to-event accuracy. Instead, they evaluated MOTOR using Brier score (Brier, 1950), 1-Calibration (D'Agostino & Nam, 2003), and two C-index variants (Harrell Jr. et al., 1996; Heagerty & Zheng, 2005). The latter only assess risk ranking[7].

Zisser & Aran (2024) presented a Transformer-based model to predict patient-specific survival probabilities for acute lymphoblastic leukemia. Although their goal was to predict the actual time to event, they again only evaluated their

---

[2]Here, a learner refers to a statistical or machine learning procedure that uses censored time-to-event data to fit a model, such as a Cox proportional hazards (CoxPH) model (Cox, 1972) or a Random Survival Forest (Ishwaran et al., 2008). Given an instance $\boldsymbol{x}_i$, the fitted model can then predict quantities such as hazard functions, survival functions, or risk scores.

[3]The *risk set* at time $t$ consists of all individuals who are still under observation and have experienced neither the event nor censoring just prior to time $t$. These individuals are considered at risk of experiencing the event at time $t$, and thus contribute to the estimation of the hazard and related quantities.

[4]Meaning $P(E > t \mid C \geq t) = P(E > t)$ for all $t$.

[5]Meaning $P(E > t \mid C \geq t, \boldsymbol{X}) = P(E > t \mid \boldsymbol{X})$ for all $t$.

[6]Real datasets can have many censored instances. For example, in a 23-year prostate cancer study, approximately 68-80% of men did not die from prostate cancer and were therefore censored (Bill-Axelson et al., 2018).

[7]A model may achieve a high C-index by correctly ranking individuals while systematically misestimating event times, *e.g.*, predicting each event time to be twice as large as its true value; rankings are preserved, giving a high C-index, but absolute errors such as MAE are large. Conversely, a model may predict event times close to the population mean for all individuals, resulting in a good (low value) MAE but poor discrimination, since little individual-level variation is captured and rankings are largely random.

model using Harrell's C-index. This was a mismatch between the objective and evaluation: they wanted correct time-to-event predictions, but the C-index only told them how good their model was at discriminating patients. Because of this, we do not know if the model's predictions can be trusted for actual time-to-event prediction.

Liu et al. (2025) presented *HACSurv*, a copula-based survival model designed to capture dependence between competing risks and censoring. Although their goal was to exploit dependencies, they evaluated their method on datasets using standard metrics, such as Antolini's C-index (Antolini et al., 2005) and the integrated Brier score (IBS), where the latter uses IPCW[8] based on a Kaplan-Meier (KM) censoring estimate (Kaplan & Meier, 1958; Graf et al., 1999; Gerds & Schumacher, 2006). Since the KM estimator assumes independent censoring, this creates a mismatch: HACSurv is designed to model dependence involving censoring, while IBS and related IPCW-based measures are only valid under the independent censoring assumption. The authors acknowledge that IBS may be unreliable under dependent censoring; nevertheless, this example illustrates the broader issue that empirical claims based on standard survival metrics require explicit justification of the censoring assumptions under which those metrics are valid.

### 3.2. Metric Requirements

Below we propose five specific desiderata for metrics that evaluate survival models. These desiderata are desirable properties and formalize what it means for an evaluation metric to be appropriate for a given scientific goal. In the following sections, we use these desiderata to analyze commonly used metrics.

**D1 Be a proper scoring rule.** A proper scoring rule is defined as a scoring function (or loss) that is minimized (or maximized, depending on convention) exactly when the predicted probabilities match the true underlying probability distribution (see Appendix C for a formal definition). A metric should be a proper scoring rule because it rewards truthful predictions – *i.e.*, models that better reflect the underlying data-generating process should receive better scores.

**D2 Be interpretable.** A metric should be expressed in domain-relevant, human-understandable units (*e.g.*,

days, weeks, months; or probabilities) and described using common, non-technical terminology. Its value should have a direct and intuitive connection to model behavior, without requiring additional transformations or assumptions. This allows practitioners and non-technical stakeholders to readily understand what is being measured and how differences in the metric reflect differences in model performance.

**D3 Be model-agnostic.** A metric should be independent of any model internals (*i.e.*, parameters) or information from the covariate distribution $P(X)$. This increases its comparability and robustness: we cannot fairly compare two models if the metric we are using depends on parameter choices or on quirks of $P(X)$. For example, a non-model-agnostic metric may assign different performance scores to two models that give identical predictions, simply because one has more parameters.

**D4 Be sensitive to miscalibration.** When a metric is used to evaluate probabilistic survival predictions, it should be sensitive to inaccurate survival probabilities. This is useful for detecting whether a model systematically overestimates or underestimates survival across the study period. For example, a model might predict that each of 100 patients has a 70% chance of surviving next year, while only 50 patients actually do.

**D5 Be robust to censoring.** A metric should support censoring in a manner consistent with its occurrence in the data. If our analysis suggests that censoring is indeed random (Definition 2.1), and censored individuals are equally likely to experience the event, the metric should reflect this observation. Conversely, if it suggests that censoring is dependent (Definition 2.3), the metric should account for this dependence.

## 4. Aspects of Evaluation and Common Metrics

We now introduce three aspects of evaluation in survival analysis and common metrics in the field; formal definitions are provided in Appendix B.

**Discrimination.** Discrimination metrics assess how well a model ranks individuals by predicted risk or event time. They measure relative ordering rather than the numerical accuracy of predicted risks, survival probabilities, or event-time estimates. Under censoring, this ordering must also be estimated under assumptions about which pairs are comparable and how censoring is handled. Figure 1 shows that the most common kind is Harrell's C-index, which estimates the proportion of correctly ordered pairs among comparable instances[9]. It can be biased under right-censoring because

---

[8]IPCW corrects for censoring by weighting observed outcomes by the inverse probability of remaining uncensored up to the relevant time. Marginal IPCW estimates the censoring survival function by $\widehat{G}(t) = \widehat{\Pr}(C > t)$, typically using the KM estimator. Conditional IPCW instead uses a covariate-dependent estimate $\widehat{G}(t \mid \boldsymbol{x}) = \widehat{\Pr}(C > t \mid \boldsymbol{X} = \boldsymbol{x})$. Using marginal IPCW implicitly assumes censoring does not depend on covariates; if censoring depends on $\boldsymbol{X}$, conditional censoring weights are required, otherwise the estimator can be biased.

[9]A pair is comparable if one instance experiences the event before the other; concordance means the instance with higher predicted risk has the event first.

*Table 1.* Overview of commonly used evaluation metrics and which desideratum each satisfies. A checkmark (✔) indicates that the metric fully satisfies the desideratum, a triangle (▲) indicates partial satisfaction, and a cross (✗) indicates that the metric does not satisfy the desideratum. For the interpretability desideratum, we rank the metrics on a scale from 1 to 7, where 1 indicates the lowest interpretability and 7 indicates the highest. The ranking follows desideratum **D2**, assigning higher ranks to metrics whose units and values are more directly understandable. The log-likelihood (LL) is included merely for comparison, as it is typically not used as a final evaluation metric.

| Desideratum | CI (Harrell) | CI (Uno) | CI (Antolini) | IBS | MAE | D-Cal | LL |
|---|---|---|---|---|---|---|---|
| **Proper scoring rule (D1)** | ✗ | ✗ | ✗ | ✔ | ▲ | ✗ | ✔ |
| **Interpretable (D2)** | #2 | #3 | #5 | #6 | #1 | #4 | #7 |
| **Model-agnostic (D3)** | ▲ | ▲ | ▲ | ✔ | ✔ | ✔ | ✗ |
| **Sensitive to miscalibration (D4)** | ✗ | ✗ | ✗ | ✔ | ✗ | ✔ | ✔ |
| **Robust to censoring (D5)** | ✗ | ▲ | ✗ | ▲ | ✗ | ✗ | ▲ |

it estimates concordance over observed comparable pairs, which need not represent all pairs defined by the true event times. Uno's C-index (Uno et al., 2011) corrects this bias using IPCW, yielding an unbiased metric under random (using KM) or independent (using a covariate-based model) censoring. However, all commonly used C-index variants remain biased under dependent censoring, and none of the C-indices are proper scoring rules when the risk score is the predicted $t$-year event probability, $\widehat{r}_i(t) = 1 - \widehat{S}(t \mid \boldsymbol{x}_i)$ (Blanche et al., 2019). They are partially model-agnostic, as they do not rely on any model internals, but do require that the model can predict individual risk scores, not just ISDs[10].

**Absolute and squared errors.** Error-based metrics evaluate numerical discrepancies between model predictions and corresponding observed outcome summaries. The Brier score (Brier, 1950) measures the accuracy of predicted survival probabilities at a given time point using squared error: the prediction is a survival probability $\widehat{S}(t \mid \boldsymbol{x}_i)$, while the observed quantity is the survival status at time $t$, $\mathbb{1}[e_i > t]$, which equals 1 if individual $i$ is event-free at $t$ and 0 otherwise. To account for censoring, it is typically estimated using IPCW and averaged over time to yield the IBS, which summarizes prediction error across the study horizon. Another commonly used error metric is the MAE, which measures the mean absolute difference between a predicted event time $\widehat{e}_i$ and the underlying event time $e_i$. Because $e_i$ is observed only when $\delta_i = 1$, MAE is either computed on uncensored observations (Schemper, 1990) or estimated using de-censoring methods such as MAE-Margin (Haider et al., 2020) or MAE-PO (Qi et al., 2023a). Although MAE and C-index may seem likely to favor the same models, this need not hold: MAE assesses accuracy of actual time predictions (*e.g.*, days, months), whereas C-index only checks rankings (Qi et al., 2023a). IBS is a proper scoring rule for survival distributions under random censoring (Rindt et al., 2022), while MAE is proper only for evaluating median survival time predictions, not full survival distributions (Qi

et al., 2023a). IBS is sensitive to miscalibration, as it penalizes inaccurate survival probabilities over time, whereas MAE is not, since it evaluates only point predictions. When censoring is handled using KM, IBS and MAE are biased under dependent censoring (Emura & Chen, 2018).

**Calibration.** Calibration metrics evaluate whether predicted survival probabilities align with the event frequencies observed in the data. A common fixed-time approach is 1-Calibration (D'Agostino & Nam, 2003), which compares predicted event or survival probabilities at a prespecified horizon $t^*$, such as $\hat{S}(t^* \mid \boldsymbol{x}_i)$, with observed event frequencies by that time. It therefore assesses calibration at one clinically relevant horizon, not over the full survival curve. D-Calibration (Haider et al., 2020) takes a different approach: rather than fixing a time point, it evaluates whether predicted survival probabilities at the event times are distributionally calibrated[11]. Thus, D-Calibration assesses calibration of the predicted ISDs, rather than calibration at a single time horizon. However, neither 1-Calibration nor D-Calibration is a proper scoring rule, since each returns a goodness-of-fit $p$-value rather than a loss to be minimized. This $p$-value is also less directly interpretable than metrics expressed in probabilities or time units. Nevertheless, D-Calibration is model-agnostic because it only requires predicted ISDs. It is sensitive to distributional miscalibration, but when censoring is handled using KM, it is not robust to dependent censoring.

## 5. Model-Metric Consistency

We introduce a guiding principle called the *Ladder Hypothesis of Model-Metric Consistency* [12] (see Figure 4). In short,

---

[10]For ISD models, risk scores can be derived in several ways: Haider et al. (2020) use the negative predicted median survival time, whereas Blanche et al. (2019) use the predicted $t$-year event probability, $\widehat{r}_i(t) = 1 - \widehat{S}(t \mid \boldsymbol{x}_i)$.

[11]Under perfect distributional calibration, the values $\widehat{S}(e_i \mid \boldsymbol{x}_i)$ for uncensored instances are uniformly distributed on $[0, 1]$. D-Calibration tests this using a goodness-of-fit test; censored observations are distributed over compatible probability intervals.

[12]As an analogy, the distortion of the model-metric ladder can be compared to the emergence of a helical structure in physics: the ladder becomes "twisted" because introducing additional constraints (*e.g.*, dependent censoring or covariate-dependent mechanisms) renders a straight alignment between model and metric assumptions statistically unstable.

a model and its evaluation metric must stand on the same rung of the assumption ladder: if they rely on different censoring assumptions, neither the reported performance nor the metric estimate can be considered reliable.

In this section, we show that this principle is often overlooked. For example, models developed under an independent censoring assumption are often evaluated using metrics designed for random censoring. Even when these limitations are known, we speculate that authors may default to off-the-shelf metrics to maintain comparability with prior work, satisfy reviewer expectations, or because no widely accepted alternative exists (see also Alternative View 2 in Section 7).

> ### Ladder Hypothesis of Model-Metric Consistency
>
> A model can be considered trustworthy only when it is evaluated with metrics whose own validity rests on the same or weaker set of statistical assumptions that also govern the model. Conversely, an evaluation metric is itself trustworthy only when any auxiliary models or estimators on which it relies satisfy those same assumptions.

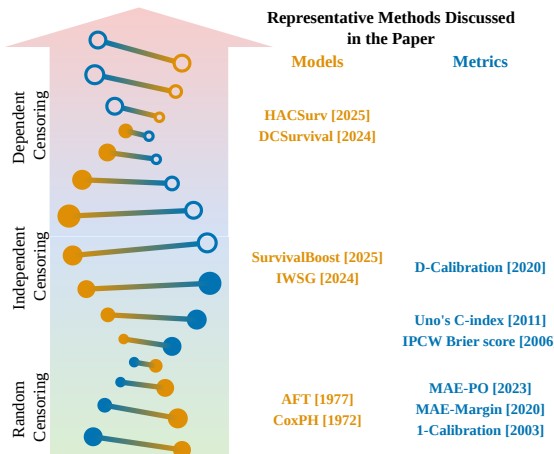

*Figure 4.* The double-helix ladder shows the current development of survival models (orange) and evaluation metrics (blue) under increasingly weakening censoring assumptions (going up). Filled dots indicate settings where valid methods exist, while hollow dots indicate unresolved gaps. The development of models has started to solve dependent censoring, while the development of metrics has not completely solved independent censoring yet. The resulting separation between the two strands highlights systematic model-metric mismatch across the assumption ladder.

### 5.1. A Simple Illustration

To illustrate the Ladder Hypothesis, we conduct a controlled experiment on synthetic data in which the predictive model

(CoxPH) is held fixed and only the censoring mechanism is varied. We generate event times from the same covariate-dependent Weibull distribution across all settings, while varying censoring across three scenarios with progressively weaker assumptions: (1) random censoring, (2) independent censoring, and (3) dependent censoring induced via a Clayton copula[13] with Kendall's $\tau \in \{0.25, 0.5, 0.75\}$, where higher Kendall's $\tau$ corresponds to stronger dependence between event and censoring times. Note that standard CI and IBS implementations (Kvamme et al., 2019; Qi et al., 2024a) rely on global censoring corrections (*e.g.*, IPCW using the KM) that do not condition on $\boldsymbol{X}$, and therefore operate under random censoring. Full details of the data-generating process and experiment design are provided in Appendix D.

We evaluate the *same* predicted survival curves using both discrimination and error-based metrics. For discrimination, we consider the oracle[14] C-index, Harrell's C-index and Uno's C-index. For error-based metrics, we consider the oracle IBS, the naive IBS (no IPCW) and the IPCW-weighted IBS. Table 2 reports the oracle performance, which reflects the model's true predictive performance under each censoring mechanism. Since the predictive model is trained under an independent censoring assumption, stronger dependent censoring can also degrade the fitted model itself; the oracle metrics therefore capture genuine changes in predictive performance rather than evaluation bias. Figure 5 reports the resulting censored-data metric errors[15]. These errors isolate the additional bias introduced by evaluating censored test data without access to the true event times. As censoring deviates from the assumptions used by the metric estimators, CI and IBS estimates become increasingly biased, and may even suggest improvement when oracle performance is degrading.

### 5.2. Random Censoring

Random censoring (*i.e.*, $E \perp\!\!\!\perp C$) is the first and most restrictive censoring assumption. Despite this limitation, almost all classical conditional survival algorithms (*i.e.*, methods that predict an individual outcome for instance $i$ given covariates $\boldsymbol{x}_i$) were derived under random censoring, including CoxPH, Accelerated Failure Time (AFT) models (Farewell & Prentice, 1977), and Random Survival Forest (Ishwaran et al., 2008), etc. Evaluation practices evolved in parallel, and thus

---

[13] A copula is a link function that encodes cross-dependency between $E$ and $C$, which applies to any marginal distribution.

[14] Oracle metrics are computed using *true* event times. This is possible in our simulation because the data-generating process records both the event time $e_i$ and censoring time $c_i$ for each instance $i$. The oracle metric therefore uses $e_i$ directly and does not rely on censoring-related assumptions.

[15] The metric error is defined as the difference between the value reported by a given censored-data metric and its corresponding oracle value computed using the true event times.

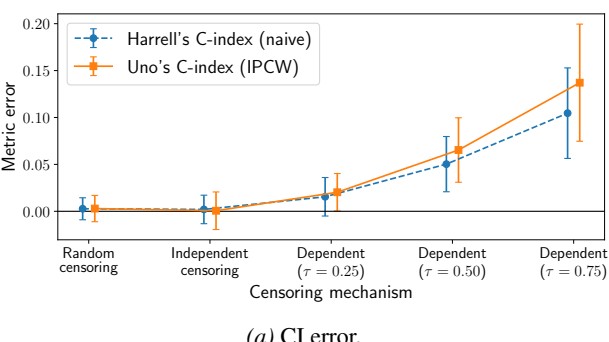

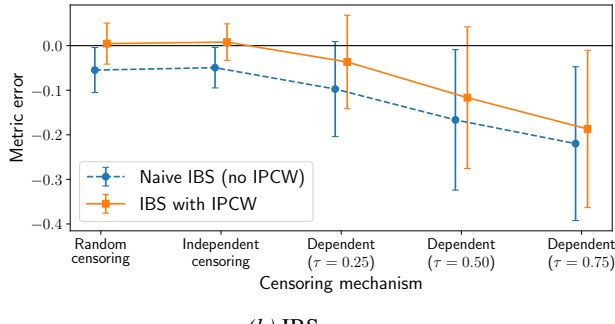

*(a)* CI error.   *(b)* IBS error.

*Figure 5.* Mean ($\pm$ SD) metric errors averaged over 100 random seeds. Errors are relative to the model's oracle performance.

*Table 2.* Oracle model performance ($\pm$ SD) under different censoring mechanisms, averaged over 100 random seeds.

| Censoring | #Events (Cens.%) | $\text{CI}_{oracle}$ $\uparrow$ | $\text{IBS}_{oracle}$ $\downarrow$ |
|---|---|---|---|
| Random | 2641 (73.6%) | 0.634 $\pm 0.018$ | 0.090 $\pm 0.040$ |
| Independent | 3157 (68.4%) | 0.634 $\pm 0.018$ | 0.084 $\pm 0.037$ |
| Dependent ($\tau = 0.25$) | 2969 (70.3%) | 0.628 $\pm 0.021$ | 0.132 $\pm 0.096$ |
| Dependent ($\tau = 0.50$) | 2758 (72.4%) | 0.618 $\pm 0.025$ | 0.199 $\pm 0.144$ |
| Dependent ($\tau = 0.75$) | 2536 (74.6%) | 0.609 $\pm 0.030$ | 0.245 $\pm 0.157$ |

many popular metrics – such as the IPCW-adjusted time-specific Brier score (Graf et al., 1999; Gerds & Schumacher, 2006), time-specific calibration (D'Agostino & Nam, 2003), and Cox-Snell residuals (Cox & Snell, 1968) – were also introduced under the random-censoring framework. When the first models were developed, these metrics had not yet been proposed; researchers instead relied on calibration-style checks, for example, comparing average predictions with the KM estimator. Since the 1980s, these advances in evaluation have led researchers to validate models built under random censoring with metrics that make the same assumption (Brier score, Cox-Snell residuals) (Ishwaran et al., 2008; Van Belle et al., 2011; Yu et al., 2011). Consequently, these models and their evaluation procedures remain *model-metric consistent*.

### 5.3. Independent Censoring

Random censoring is rarely justifiable in practice; instead, modern survival analysis typically assumes that the censoring time $C$ may depend on covariates $X$ but is conditionally independent of the event time $E$ given $X$ (*i.e.*, $E \perp\!\!\!\perp C \mid X$).

When independence holds only *given $X$*, naively treating data as randomly censored can inflate or attenuate risk esti-

mates. As an example, patients who leave follow-up early may differ systematically (*e.g.*, they are younger, healthier, wealthier) from those who remain, skewing both model training and evaluation. Ignoring this bias can yield biased estimates and misleading clinical conclusions. To mitigate censoring-induced bias, recent methods explicitly account for the censoring mechanism, either by modeling it directly or by reweighting censored observations (Robins & Rotnitzky, 1992; Cole & Hernán, 2004; Chapfuwa et al., 2021; Han et al., 2021; 2022; Alberge et al., 2025). Examples include the inverse-weighted survival game (IWSG) of Han et al. (2021) and SurvivalBoost (Alberge et al., 2025), both of which exploit covariate information in $X$ to adjust for censoring.

Meanwhile, many popular evaluation metrics can differ from their oracle counterparts even under random censoring, especially in their commonly used unweighted forms, such as Harrell's C-index and the uncorrected Brier score (see Figure 5). This motivates IPCW-variants (see Section 4) that are unbiased if censoring weights are correctly estimated. Unfortunately, some papers still report *unweighted* C-index or Brier scores (see Figure 1); the resulting model-metric mismatch reintroduces the very same bias those models aim to eliminate. To our knowledge, consistently estimating IPCW weights under independent censoring is nontrivial in practice without additional modeling assumptions. As a result, evaluation procedures based on IPCW may fail to be consistent in practice. Therefore, these models and their evaluation procedures are not *model-metric consistent*.

### 5.4. Dependent Censoring

Dependent censoring (*i.e.*, $E \not\perp\!\!\!\perp C \mid X$) is the most complicated form of censoring because the joint law of $(E, C)$ is no longer identifiable from the observed-data distribution $\Pr(X, T, \Delta)$ without further structural assumptions (Tsiatis, 1975). Existing approaches include sensitivity analyses for untestable dependence assumptions (Siannis et al., 2005; Huang & Zhang, 2008), partial-identification bounds under weaker assumptions, and model-based corrections that im-

pose structure on the joint distribution, for example through copulas (Huang & Zhang, 2008; Emura & Chen, 2018; Foomani et al., 2023; Zhang et al., 2024a; Liu et al., 2025) or proximal causal inference (Tchetgen Tchetgen et al., 2024; Ying, 2024). Representative copula-based approaches include DCSurvival (Zhang et al., 2024a) and HACSurv (Liu et al., 2025), which explicitly model dependence using a copula function. These developments mainly concern model estimation, however, not model evaluation.

Unfortunately, virtually all commonly used performance measures (Harrell's C-index, Uno's C-index, Brier score, MAE, D-Calibration) assume at least independent censoring. Under dependent censoring, they become biased, breaking model-metric consistency. Because there are no broadly accepted assumption-aligned metrics, most papers tackling dependent censoring report results on synthetic data where the ground truth is known (Qi et al., 2023a).Worryingly, some studies (see the last example in Section 3.1) nonetheless declare state-of-the-art performance on real datasets with alleged dependent censoring using independent-censoring metrics such as Antolini's C-index, IBS, or D-Calibration. Such scores are biased, rendering any claimed superiority an artifact of model-metric mismatch. One recent line of work by Lillelund et al. (2025a) addresses this gap by replacing the KM estimator used inside standard survival metrics with the Copula-Graphic estimator (Zheng & Klein, 1995). This estimator jointly models event and censoring times under an Archimedean copula (Emura & Chen, 2018), and yields copula-based analogues of common metrics that can empirically recover model error better under dependent censoring. However, such approaches require specifying a copula family, which introduces a strong and typically unverifiable assumption. Moreover, existing evidence is still mainly empirical, with few theoretical guarantees such as consistency or asymptotic normality for the resulting metrics.

In summary, survival analysis lacks broadly accepted evaluation metrics that remain unbiased under dependent censoring. Until such tools are developed and validated, performance claims for state-of-the-art models in this setting are, at best, speculative in real datasets. **Our central message is therefore a plea for recalibration:** researchers must scrutinize the assumptions behind the metrics they report, and the community should prioritize creating assumption-aligned evaluation procedures before proclaiming new modeling breakthroughs. This includes developing evaluation procedures that are robust to dependent censoring. In our opinion, rigorous evaluation should be the next frontier for progress under dependent censoring.

## 6. Practical Recommendations

Below we provide some practical recommendations for evaluation and consistency in survival analysis.

**Recommendation 1 (Stick to your goal):** In application-oriented studies, the primary evaluation metric should follow directly from the study objective, prediction target, and evaluation aspect: discrimination, absolute or squared error, calibration, or another aspect (Section 4). Risk ranking objectives (*e.g.*, prioritizing patients for a critical medical resource) call for discrimination metrics such as C-index or AUC variants (Hung & Chiang, 2010); time-to-event estimation favors error-based metrics such as MAE or MSE; and decision-making based on predicted survival probabilities is best assessed using calibration-oriented measures. Finally, among metrics that address the same aspect, the desiderata in Section 3.2 help clarify what the trade-offs are.

**Recommendation 2 (Evaluate from multiple angles):** In methodology-oriented studies (*e.g.*, developing a new general-purpose algorithm), we recommend reporting at least one representative metric from each of the aforementioned aspects (Section 4). Many new deep-learning methods increasingly predict the full survival curve (see Section 2.1); however, because the true survival curve is unobserved and its tail is often unreliable due to censoring, relying only on metrics that assess the full survival function can be misleading. We also recommend evaluating well-chosen scalar summaries[16], which can provide more stable and practically relevant feedback than comparing entire survival curves. For example, clinicians managing kidney disease usually act on a 2-year or 5-year predicted risk score (Stevens et al., 2024).

**Recommendation 3 (Strive for consistency):** Reported performance should be interpreted relative to the censoring assumptions supported by the chosen metric. Survival metrics encode specific assumptions about censoring, observability, and what constitutes a good prediction (see Section 4). We therefore recommend explicitly stating one's assumptions about censoring, choose metrics that align with these assumptions (*e.g.*, estimate censoring weights in Uno's C-index using a CoxPH model) and openly declare when model-metric consistency cannot be guaranteed. When such alignment is not possible, robustness checks, sensitivity analyses, or complementary metrics should be used to show that conclusions are not artifacts of the evaluation procedure itself. As alluded to, under conditionally independent censoring, a practical path forward is to replace marginal censoring corrections with covariate-dependent ones[17].

---

[16]Examples include landmark event probabilities $1 - \widehat{S}(t^* \mid \boldsymbol{x}_i)$, the predicted median survival time $\inf\{t : \widehat{S}(t \mid \boldsymbol{x}_i) \leq 0.5\}$, and the restricted mean survival time (RMST) $\int_0^{t^*} \widehat{S}(t \mid \boldsymbol{x}_i)\,\mathrm{d}t$. The mean survival time is given by $\int_0^\infty \widehat{S}(t \mid \boldsymbol{x}_i)\,\mathrm{d}t$, but is often less stable because it depends on the tail of the predicted survival curve.

[17]For IPCW-based metrics, this means estimating the censoring survival function conditionally, $\widehat{G}(t \mid \boldsymbol{x}_i) = \widehat{\Pr}(C > t \mid \boldsymbol{X} = \boldsymbol{x}_i)$, rather than using a marginal KM estimate $\widehat{G}(t)$.

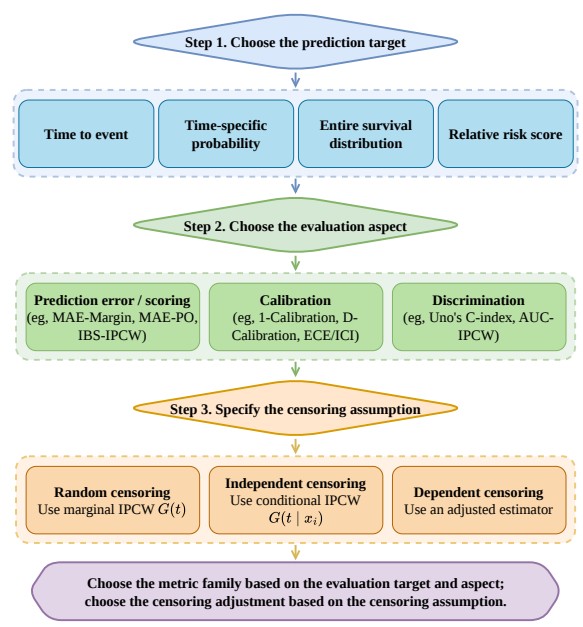

*Figure 6.* A workflow for selecting survival prediction metrics.

To make these recommendations actionable, we summarize them in the practical checklist (shown in Figure 6), which guides the selection of evaluation metrics based on the prediction target, evaluation aspect and censoring assumption.

## 7. Alternative Views

**Alternative View 1** *In methodological papers, evaluation is often intended to be application-agnostic, and the C-index is therefore used as a general-purpose measure of predictive performance. Because it assesses a model's ability to correctly order individuals by risk without requiring well-calibrated absolute probabilities, it is robust to differences in baseline hazard specification and outcome prevalence. Empirically, improvements in C-index are frequently observed to coincide with improvements in other performance measures, making it a practical primary metric for comparing methods.*

**Response** Strong ranking performance does not imply calibrated survival probabilities or accurate time-to-event estimates (Hartman et al., 2023; Qi et al., 2023a). This is likely more pronounced for models that are trained only to optimize a ranking loss, *e.g.*, DeepSurv (Katzman et al., 2018) and the Hierarchical model (Tjandra et al., 2021), which have been shown to perform poorly on error and calibration-based metrics when reproduced (Lillelund et al., 2026a). We believe methodological contributions should be evaluated broadly rather than selectively, with metrics that cover the main as-

pects of performance, so that a method's strengths, weaknesses, and relevant baselines are clearly visible. Such transparency is essential for understanding how and when the proposed method offers real advantages.

**Alternative View 2** *Even if standard evaluation metrics are biased under censoring, their use is still justified in practice. Authors may default to standard metrics to maintain comparability with prior work, to satisfy reviewer expectations or because no widely accepted alternatives exist under weaker censoring assumptions. Using familiar metrics also facilitates communication and interpretation within the community. Moreover, when comparing two models $M_A(\cdot)$ and $M_B(\cdot)$ on the same dataset, any censoring-induced bias from the dataset is shared across models, so relative performance comparisons remain meaningful.*

**Response** First, comparability with prior work does not mitigate the risk of propagating systematic error: consistent use of biased metrics propagates bias, not just noise. Biased metrics can make apparent progress an artifact of evaluation. Second, censoring-induced bias is not necessarily shared equally across models, especially when models rely on different censoring assumptions. For example, if $M_A(\cdot)$ implicitly assumes independent censoring while $M_B(\cdot)$ is designed to model dependent censoring, then evaluating both with a metric that assumes independence may fail to credit $M_B(\cdot)$ appropriately for its better predictive performance. Third, if no unbiased alternatives exist, that does not justify making strong empirical performance claims; instead, it means that some claims simply cannot yet be made reliably, especially when a method explicitly targets dependent censoring (see Section 3.1).

## 8. Conclusion

We argue that a substantial number of papers in survival analysis evaluate models using metrics that do not reflect the stated scientific goal or implicitly rely on censoring assumptions that are rarely stated, justified, or corrected for in practice. In a survey of 92 papers published between 2023 and 2025, we found approximately 72% fall under our criteria of misalignment. This suggests that this issue is widespread, and unfortunately can lead to overstated performance claims or prevent new models from being credited for methodological breakthroughs in modeling techniques. Therefore, we gave five desiderata to guide metric selection and showed why model-metric alignment is important with a concrete experiment. Going forward, we advocate for evaluation protocols in which metrics, assumptions, and data-generating mechanisms are made explicit and ideally aligned, and for further research on new metrics, especially under dependent censoring.

## Acknowledgements

This research received support from the Natural Science and Engineering Research Council of Canada (NSERC), the Canadian Institute for Advanced Research (CIFAR), and the Alberta Machine Intelligence Institute (Amii).

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

# A. Structured Survey

We conducted a structured survey to examine how evaluation metrics are used in recent survival analysis publications. Our review covered papers published between 2023.01.01 and 2025.12.31 across major scientific databases, including Google Scholar, ScienceDirect, Scopus, Springer Nature, and IEEE Xplore. We supplemented this database search with manual screening of major machine-learning venues (*e.g.*, NeurIPS, ICML, ICLR, AISTATS), selected biomedical conferences (*e.g.*, EMBC), and relevant journals. To identify relevant articles, we searched for combinations of survival-analysis and evaluation-related keywords (see Table 3). Papers were included if they addressed survival analysis using a learned predictive model and contained an empirical evaluation section; purely methodological statistics papers and epidemiological studies without predictive modeling were excluded. For each paper, we extracted the stated problem type, modeling objectives, the metrics used in the experimental section, and whether the chosen metrics aligned with the stated goals and with the censoring assumptions of the data. Our sampling strategy intentionally focuses on machine learning-oriented survival analysis and may therefore over-represent ML and biomedical venues relative to traditional biostatistics.

*Table 3.* Keywords used in the structured survey for identifying survival analysis papers with explicit evaluation protocols.

| Aspect | Keywords |
|---|---|
| General | survival analysis, time-to-event, censoring, evaluation |
| Metrics (Discrimination) | concordance index, c-index, auc, harrell, uno, antolini |
| Metrics (Absolute and squared errors) | brier score, ibs, mae, mse |
| Metrics (Calibration) | calibration, d-calibration, ici, ece |

Table 4 summarizes the frequency with which evaluation metrics appear across the articles included in our survey. These counts form the basis of the analysis presented in Figure 1.

*Table 4.* Metrics used in survival analysis papers (2023-2025). Total number of included articles: 92.

| Evaluation Metric | Number of Papers (%) | Reference |
|---|---|---|
| CI (Harrell) | 62 (67.4%) | (Pomsuwan & Freitas, 2024; Nikolaou et al., 2025; Lillelund et al., 2025b; Zhang et al., 2025; Abuhantash et al., 2025; Huang et al., 2024; Li et al., 2025; Birolo et al., 2025; Tang et al., 2026; Zhou et al., 2025; Xu et al., 2024; Hernández–Pérez et al., 2025; Qi et al., 2024c; Zhang et al., 2024b; Knottenbelt et al., 2025; Xu & Guo, 2023; Luo et al., 2023; Hou et al., 2023; Cui et al., 2024b; Gupta et al., 2023; Cui et al., 2024a; Vauvelle et al., 2023; Ghosh et al., 2025; Tang et al., 2023; Cui et al., 2025; Haredasht et al., 2023; Do et al., 2023; Liu et al., 2024; Bone-Winkel & Reichenbach, 2024; Kim, 2025; Zhang et al., 2026; Wang et al., 2024; Baniecki et al., 2025; Wu et al., 2024; Lillelund et al., 2026a; Wang et al., 2025a; Song et al., 2024; Monod et al., 2026; Abdallah et al., 2024; Rechichi et al., 2024; Lillelund et al., 2026b; Zhang et al., 2024c; Xie et al., 2026; Yan et al., 2024; Fan et al., 2024; Buyrukoğlu, 2024; Rahman et al., 2023; Supriya & Anitha, 2024; Woodward et al., 2024; Norcliffe et al., 2023; Rollo et al., 2024; Tan et al., 2023; Chen et al., 2025; Luo et al., 2025; Hajianfar et al., 2023; Lee et al., 2024; Qi et al., 2024b; Zisser & Aran, 2024; Lillelund et al., 2023; Qi et al., 2023b; Park et al., 2026; Seidi et al., 2024) |
| CI (Uno) | 7 (7.6%) | (Lillelund et al., 2025b; Pu et al., 2025; Archetti & Matteucci, 2023; Bone-Winkel & Reichenbach, 2024; Lillelund et al., 2026b; Sluiskes et al., 2024; Lillelund et al., 2023) |

| Evaluation Metric | Number of Papers (%) | Reference |
|---|---|---|
| CI (Antolini) | 13 (14.1%) | (Birolo et al., 2025; Yang & Qiu, 2025; Shen & Chen, 2026; Ogutu et al., 2025; Wan et al., 2024; Bleistein et al., 2024; Lillelund et al., 2025c; Hu & Chen, 2024; Apellániz et al., 2024; Zhao et al., 2024; Chen, 2024; Massoua et al., 2024; Tang et al., 2025) |
| TD-AUC | 18 (19.6%) | (Zhang et al., 2025; Li et al., 2025; Birolo et al., 2025; Hernández–Pérez et al., 2025; Pu et al., 2025; Wan et al., 2024; Ghosh et al., 2025; Wang et al., 2025b; Baniecki et al., 2025; Van Ness et al., 2023; Lu et al., 2023; Blythe et al., 2024; Yan et al., 2024; Kim, 2023; Rollo et al., 2024; Tan et al., 2023; Luo et al., 2025; Massoua et al., 2024) |
| Brier score | 10 (10.9%) | (Huang et al., 2024; Wan et al., 2024; Do et al., 2023; Baniecki et al., 2025; Lu et al., 2023; Buyrukoğlu, 2024; Woodward et al., 2024; Norcliffe et al., 2023; Tan et al., 2023; Luo et al., 2025) |
| Brier score (IPCW) | 26 (28.3%) | (De Bin & Stikbakke, 2023; Lillelund et al., 2025b; Birolo et al., 2025; Pu et al., 2025; Qi et al., 2024c; Cui et al., 2024b; Shen & Chen, 2026; Ogutu et al., 2025; Cui et al., 2024a; Lillelund et al., 2025c; Hu & Chen, 2024; Liu et al., 2024; Archetti & Matteucci, 2023; Van Ness et al., 2023; Apellániz et al., 2024; Lillelund et al., 2026a; Wang et al., 2025a; Steinberg et al., 2024; Wu et al., 2023; Monod et al., 2026; Lillelund et al., 2026b; Zhao et al., 2024; Lee et al., 2024; Qi et al., 2024b; Lillelund et al., 2023; Park et al., 2026) |
| MAE (Uncens.) | 7 (7.6%) | (Zhang et al., 2025; Qi et al., 2023a; Huh et al., 2025; Yang & Qiu, 2025; Kim, 2025; Wang et al., 2025a; Zisser & Aran, 2024) |
| MAE (Cens.) | 9 (9.8%) | (Lillelund et al., 2025b; Zhang et al., 2025; Qi et al., 2023a; 2024c; Lillelund et al., 2025c; 2026a;b; Qi et al., 2024b; 2023b) |
| MSE/RMSE | 5 (5.4%) | (Huh et al., 2025; Kim, 2025; Wang et al., 2025a; Sluiskes et al., 2024; Curth & van der Schaar, 2023) |
| ECE/ICI | 2 (2.2%) | (Lillelund et al., 2025c; Do et al., 2023) |
| D-Calibration | 9 (9.8%) | (Lillelund et al., 2025b; Qi et al., 2024c; Lillelund et al., 2025c; 2026a; Monod et al., 2026; Lillelund et al., 2026b; Lee et al., 2024; Qi et al., 2023b; Park et al., 2026) |
| 1-Calibration | 1 (1.1%) | (Steinberg et al., 2024) |
| None | 4 (4.3%) | (Feng et al., 2026; Bian et al., 2025; Gögl et al., 2026; Frauen et al., 2026) |

Table 5 provides a detailed qualitative evaluation of each paper included in our structured review. These counts form the basis of the analysis presented in Figure 2. For every work, we assess whether the evaluation metrics align with the stated modeling objectives and whether the censoring assumptions required by those metrics are respected, ignored, or implicitly violated.

To assign the labels **Yes**, **Partially**, and **No**, we apply the following criteria:

- **A paper is labeled Yes** if it satisfies *both*:
    - metrics used are appropriate for the stated objectives (*e.g.*, risk ranking, time-to-event prediction, calibration), and
    - censoring is stated, justified, or actually corrected for (*e.g.*, using IPCW with KM for random censoring).

- **A paper is labeled Partially** if it satisfies *exactly one* of the above:

  – objectives are evaluated using appropriate metrics but censoring bias is not addressed, *or*

  – censoring is handled correctly but key metrics for the stated objectives are missing.

- **A paper is labeled No** if it satisfies *neither* criterion. Typical examples include:

  – claiming to predict time-to-event outcomes while reporting only Harrell's C-index, or

  – using metrics that are invalid under censoring without correction, justification or acknowledging of the bias.

We consider the following classes of metrics appropriate for modern survival analysis:

- **Discrimination:** Harrell, Uno, and Antolini C-indices; time-dependent AUC (TD-AUC).

- **Absolute and squared errors:** Brier score, Brier score (IPCW), MAE/RMSE, and censored variants.

- **Calibration:** ECE/ICI, D-Calibration, 1-Calibration.

A paper's score is decremented when their stated goals (*e.g.*, predicting survival curves, estimating event times, or providing calibrated risk scores) are not evaluated using appropriate metrics, or when censoring renders the chosen metrics invalid without appropriate adjustment or acknowledgment. These criteria allow us to assess whether each paper's evaluation methodology genuinely supports its claims.

*Table 5.* Structured survey of survival analysis papers published between 2023-2025. Each entry includes the problem type, stated objectives, evaluation metrics used, and an assessment of whether the chosen metrics align with stated goals and censoring assumptions.

| Paper | Problem Type | Objectives | Correctly Evaluated? | Explanation |
|---|---|---|---|---|
| (Abdallah et al., 2024) | Methodology | Predict time-to-event outcomes (disease progression, relapse, etc.), compare DBN performance against existing survival models, emphasize ranking / discrimination ability. | No | Wrong metrics for the stated TTE objective, censoring bias not addressed. |
| (Abuhantash et al., 2025) | Application | Predict time to conversion from CN to MCI, predict risk prediction / progression modeling, and clinical usefulness (risk stratification, early detection, etc.). | No | Missing TTE metrics, heavy censoring with unadjusted Harrell's CI and no discussion of assumptions. |
| (Apellániz et al., 2024) | Methodology | Estimate the full predictive distribution of time-to-event, handle right-censored data, compare survival models using standard survival metrics. | Yes | They evaluate discrimination and calibration, and address censoring by using IPCW. |
| (Archetti & Matteucci, 2023) | Methodology | Provide an efficient, federated survival model, achieve strong discrimination performance, achieve good calibration of survival probabilities, and handle censored, incomplete data across distributed clients. | Partially | Missing TTE metrics or actual calibration metrics (*e.g.*, D-calibration), but they correct for random censoring. |
| (Baniecki et al., 2025) | Methodology | Introduce time-dependent interpretability methods for survival analysis, demonstrate the usefulness of these interpretability tools on: Hospital Length of Stay (LoS) modeled as a time-to-event prediction problem, and Cancer survival prediction using TCGA multi-omics data. | No | Metrics do not match TTE or calibration objectives, censoring bias not addressed. |
| (Bian et al., 2025) | Methodology | Introduce nonparametric boosting algorithms for regression and classification when outcomes are interval-censored, using censoring-unbiased transformations within functional gradient descent to achieve accurate and scalable prediction under interval censoring. | No | No survival metrics used in real-world dataset. |
| (Birolo et al., 2025) | Methodology | Evaluate predictive performance of survival methods under PH / non-PH / linear / nonlinear conditions, compare survival metrics and show when Harrell CI is inappropriate, assess calibration, study sample-size effects and computational cost. | Partially | Metrics align with the goals, but censoring bias not addressed. |

| Paper | Problem Type | Objectives | Correctly Evaluated? | Explanation |
|---|---|---|---|---|
| (Bleistein et al., 2024) | Methodology | Model dynamic survival processes from longitudinal covariates by learning a controlled latent-state representation that drives the event intensity over time; goal is improved individualized risk ranking / time-varying risk prediction from irregularly sampled histories. | Yes | Metrics match their goals and censoring bias addressed. |
| (Blythe et al., 2024) | Application | Use a Cox time-to-event model to improve ranking of patients by risk, compare ranking performance of Cox vs discrete-time logistic regression, develop a model useful for triage prioritization. | Yes | Their only concern is ranking and Uno's AUC handles random censoring properly. |
| (Bone-Winkel & Reichenbach, 2024) | Application | Estimate hazard ratios and then assign borrowers into risk-rating groups, evaluate whether ratings separate default risk, IRR, and survival curves. | Yes | Metrics match risk prediction goal, and they correct for random censoring using Uno's CI. |
| (Buyrukoğlu, 2024) | Application | Predict time-to-relapse in breast cancer, compare the predictive ability of CoxPH vs. RSF vs. CForest, evaluate model discrimination and prediction error, provide variable importance explanations, illustrate survival probability curves for hypothetical patients. | No | Missing TTE metrics and censoring assumptions not addressed. |
| (Chen, 2024) | Methodology | Develop a deep kernel survival model (survival kernet) that scales to large datasets while remaining interpretable, and provide finite-sample accuracy guarantees for the predicted individual survival distributions. | Partially | Metrics match their ranking goals, but censoring bias not addressed. |
| (Chen et al., 2025) | Application | Develop a new multimodal survival model (TDMFS) using Tucker decomposition, improve survival prediction accuracy (*i.e.*, risk prediction, not explicit time-to-event estimation), perform risk stratification with Kaplan–Meier curves to show separation of high- vs. low-risk groups. | Partially | Proper metrics for risk ranking, but censoring handled poorly; no calibration; not suitable for true time-to-event evaluation. |
| (Cui et al., 2024b) | Methodology | Improve survival prediction performance for glioma patients, predict survival risk / rank patients, produce more accurate survival curves (they include predicted KM-like curves), compare their model to baseline models on multiple metrics. | No | Missing metrics for calibration and survival-curve accuracy. Harrell CI used without assumptions; classification metrics invalid under censoring. |
| (Cui et al., 2024a) | Methodology | Improve survival prediction accuracy, improve risk discrimination, improve calibration, handle censored data more effectively via contrastive learning, provide better patient-level survival curves (CIF). | Partially | Missing TTE error metrics, incomplete calibration, but censoring addressed under the random assumption. |
| (Cui et al., 2025) | Application | Predict patient-specific survival probabilities over time, model survival curves and hazard functions, assess model discrimination (ranking), provide interpretability (SHAP). | No | They want to evaluate survival probabilities, but use only Harrell CI. No use of censoring-robust metrics. |
| (Curth & van der Schaar, 2023) | Methodology | Analyze how competing events induce covariate shift for different causal estimands. | Yes | RMSE of estimated HTEs is the correct metric for their causal-identification goals. |
| (De Bin & Stikbakke, 2023) | Methodology | Develop non-PH model to predict first hitting time, evaluate predictive performance and calibration, no ranking. | Partially | IBS with IPCW is appropriate, but calibration only evaluated with plots. |
| (Do et al., 2023) | Methodology | Ensure fairness of survival-time predictions across sensitive groups, measure disparities in survival prediction. | Partially | Metrics are correct for their fairness objective, but Harrell's CI and BS are not corrected for censoring. |
| (Fan et al., 2024) | Methodology | Build a stable Cox model that improves generalization of risk ranking under distribution shifts, identify stable variables that maintain predictive value across cohorts, evaluate prognostic performance on real-world cancer datasets. | Partially | Metrics match risk prediction goal, but censoring bias not addressed. |
| (Feng et al., 2026) | Methodology | Develop a reinforcement-learning framework for alternating recurrent-event survival data that optimizes a probability-based objective (maximize the probability inter-event durations exceed a clinically meaningful threshold) rather than expected outcomes, enabling standard RL training via a lower-bound reformulation. | No | No survival metrics used in real-world dataset. |

| Paper | Problem Type | Objectives | Correctly Evaluated? | Explanation |
|---|---|---|---|---|
| (Frauen et al., 2026) | Methodology | Provide a toolbox of Neyman-orthogonal/meta-learning survival learners for estimating heterogeneous treatment effects from censored time-to-event data, improving robustness (e.g., under low overlap) and allowing plug-in of arbitrary base learners. | No | No survival metrics used in real-world dataset. |
| (Ghosh et al., 2025) | Application | Predict the time to clinical progression from CN to MCI/AD, early risk stratification using longitudinal biomarkers. | No | Missing TTE metrics, censoring bias not addressed. |
| (Gupta et al., 2023) | Methodology | Estimate ATE/ITE under censoring; reduce confounding; improve causal accuracy. | Partially | Causal metrics are correct, but the censoring bias is not addressed. |
| (Gögl et al., 2026) | Methodology | Estimate heterogeneous, dosage-dependent treatment effects for time-to-event outcomes by learning personalized survival predictions under continuous-valued interventions (dose-response), supporting individualized clinical decision-making from observational TTE data. | No | No survival metrics used in real-world dataset. |
| (Hajianfar et al., 2023) | Application | Predict time-to-event OS using radiomic + clinical features, identify strong preprocessing / feature selection / ML combinations, stratify patients into risk groups (Kaplan–Meier used for validation). | Partially | Harrell's CI measures risk ranking, but no IBS, MAE, RMSE, or other time-error metrics are reported. |
| (Haredasht et al., 2023) | Methodology | Predict time-to-event (survival time) and improve accuracy using a semi-supervised approach that uses censored observations more effectively, develop a self-training framework (STUART) that augments training data with confident predictions on censored instances, compare predictive performance against RSF, Cox-based models, and SP-AFT. | No | Missing TTE metrics and only use Harrell CI. No calibration tests and censoring bias left unaddressed. |
| (Hernández–Pérez et al., 2025) | Application | Evaluate and compare survival analysis models for melanoma prognosis, predict survival outcomes (OS, SS, DFS) for melanoma, assess both predictive performance and interpretability. | Partially | They use correct metrics for their ranking and calibration objectives, but censoring not properly handled (*e.g.*, no IPCW). |
| (Hou et al., 2023) | Methodology | Perform time-to-event prediction, perform survival clustering using interpretable expert distributions, compete with state-of-the-art deep methods (Deep Cox, DSM, SCA, VaDeSC, show that the mixture-of-experts formulation improves interpretability. | No | Missing TTE metrics and calibration, censoring bias is not addressed. |
| (Hu & Chen, 2024) | Methodology | Improve fairness across all sufficiently large subpopulations using a DRO-based method, be applicable to any survival model, including Cox, DeepHit, SO-DEN, maintain accuracy (discrimination & risk prediction quality), handle censoring within fairness metrics. | Yes | Metrics match stated goals and censoring handled appropriately. |
| (Huang et al., 2024) | Methodology | Predict continuous time-to-event values, quantify uncertainty using belief functions & random fuzzy numbers, handle right censoring correctly via modified evidential likelihood, achieve good predictive performance, achieve good calibration. | No | Missing TTE accuracy metrics and censoring bias not addressed. |
| (Huh et al., 2025) | Methodology | Jointly model longitudinal + TTE data; propagate uncertainty from both sources; improve risk prediction accuracy; quantify prediction uncertainty. | Partially | Metrics cover TTE accuracy but not calibration; censoring assumptions not addressed; no formal calibration test. |
| (Kim, 2025) | Methodology | Predict survival outcomes (vital status), estimate Heterogeneous Treatment Effects (HTE), evaluate differences in treatment effects across race, improve prediction with HT and IPTW weighting. | No | Metrics do not match their goals, and censoring bias ignored in evaluation. |
| (Kim, 2023) | Methodology | The paper proposed a novel Bayesian survival model to estimate a hazard function as a nonlinear regression function of covariates from survival data with time-varying covariates. | Yes | Metrics match goals and censoring bias addressed. |

| Paper | Problem Type | Objectives | Correctly Evaluated? | Explanation |
|---|---|---|---|---|
| (Knottenbelt et al., 2025) | Methodology | Predict log-partial hazard (risk ranking), recover symbolic formulae representing hazard contributions, automatic feature pruning using KAN structure. | Partially | They use correct metric for risk ranking, but censoring bias is not addressed. |
| (Lee et al., 2024) | Methodology | Improve discrimination (risk ranking) in deep survival models without sacrificing calibration by using survival-outcome-aware contrastive learning (paired with likelihood training) so that patients with similar event times are embedded closer while keeping well-calibrated survival probabilities. | Yes | They claim to address discrimination and calibration and evaluate both thoroughly; censoring assumptions stated and handled. |
| (Li et al., 2025) | Application | Predict RFS, a time-to-event outcome (RFS = time from surgery to recurrence/death), predict intracranial recurrence at fixed horizons (6, 12, 18 months), risk stratification using survival curves, construct a prognostic nomogram. | No | Metrics do not match TTE objective, censoring handling is incorrect for chosen metric. |
| (Lillelund et al., 2025b) | Application | Predict functional decline as an event in ALS patients, perform risk ranking, make counterfactual predictions. | Yes | Correct metrics for objectives and censoring bias addressed. |
| (Lillelund et al., 2025c) | Methodology | Compare probabilistic training approaches (VI vs. MCD vs. SNGP), evaluate prediction performance (ranking + time-to-event accuracy), evaluate calibration performance (distribution, coverage, probability calibration), provide computational efficiency comparisons. | Partially | Uses the appropriate metrics for ranking, time-to-event prediction, and calibration, but censoring bias not addressed. |
| (Lillelund et al., 2026a) | Methodology | Model multiple time-to-event distributions jointly, improve event ordering (local survival discrimination), improve global discrimination, provide population survival curves. | Yes | Correct metrics for objectives and censoring bias addressed. |
| (Lillelund et al., 2026b) | Application | Predict individual fall risk within 6 months, provide a ranking of individuals, predict time-to-event, provide calibrated survival curves for decision support. | Yes | Uses the appropriate metrics for ranking, time-to-event prediction, and calibration. Censoring bias addressed. |
| (Lillelund et al., 2023) | Methodology | Assess whether Bayesian NNs improve survival prediction under Cox PH, quantify uncertainty, compare deterministic vs Bayesian models on predictive performance. | Yes | Metrics align with objectives and appropriately handle censoring. |
| (Liu et al., 2024) | Methodology | Speed up and stabilize training of Cox proportional hazards models on modern high-dimensional data by introducing globally convergent surrogate-optimization methods, enabling practical training of sparse/high-quality Cox models and other Cox-related extensions. | Partially | Metrics match objective, but censoring assumptions never stated. |
| (Lu et al., 2023) | Methodology | Predict time-to-event outcomes, generate individualized survival curves, identify risk factors, compare RSF to KM. | Partially | Metrics do not fully address TTE prediction, and censoring bias is not addressed. |
| (Luo et al., 2023) | Methodology | Predict risk; evaluate feature selection; model uncertainty. | No | Only partial alignment with objectives and censoring bias is not addressed. |
| (Luo et al., 2025) | Application | Develop and validate machine-learning-based survival prediction models for overall survival (OS) in SP-NSCLC, predict survival probabilities over time, evaluate time-dependent predictor importance via permutation explanation, risk stratification into high- and low-risk groups via KM curves. | Partially | They use correct metrics for ranking goal, but censoring bias not addressed (no IPCW) |
| (Massoua et al., 2024) | Methodology | Predict the actual time-to-event (event-time) distribution, estimate survival density and survival function, improve accuracy of time-to-event estimation under noise. | No | Metrics do not match TTE objective, censoring bias not addressed. |
| (Monod et al., 2026) | Methodology | Provide deep survival prediction with principled Bayesian uncertainty quantification to produce well-calibrated survival functions and uncertainty estimates (especially in data-scarce regimes) while matching or improving discriminative performance. | Yes | They claim to address discrimination and calibration and evaluate both thoroughly, censoring assumptions stated and handled. |

| Paper | Problem Type | Objectives | Correctly Evaluated? | Explanation |
|---|---|---|---|---|
| (Nikolaou et al., 2025) | Application | Compare multimodal vs unimodal survival models for overall survival (OS) on TCGA; main focus on improved discrimination / predictive performance and multimodal fusion strategy (late fusion). | Partially | They use correct metric for risk ranking and acknowledges censoring-induced bias, but still uses a censoring-biased metric. |
| (Norcliffe et al., 2023) | Methodology | Formalize synthetic survival data generation, generate time-to-event data that incorporates censored and uncensored data, evaluate quality of generated data. | Partially | Missing TTE metrics, but censoring stated as assumed random. |
| (Ogutu et al., 2025) | Application | Apply survival models to predict time to HIV infection using high-dimensional data with time-varying cytokines. | Yes | Correct metrics for objectives and censoring bias addressed. |
| (Park et al., 2026) | Methodology | Develop a new calibration method for survival models (KSP), improve calibration across the entire survival distribution without harming discrimination, provide theoretical foundations for KS-cal and show consistency, empirically evaluate calibration performance across datasets and survival models. | Yes | Metrics align with claims and censoring assumptions addressed. |
| (Pomsuwan & Freitas, 2024) | Methodology | Predict time-to-event outcomes, cope with censored data, automatically select the best survival model + hyperparameters, improve predictive accuracy over baseline methods, evaluate performance across multiple datasets. | No | Missing TTE metrics and censoring bias not addressed. |
| (Pu et al., 2025) | Application | Predict 5-year cause-specific survival (CSS) for cervical cancer patients, compare five survival models, identify important clinical predictors of CSS using SHAP, provide risk stratification (KM curves + log-rank test). | Yes | Metrics match objectives and censoring addressed. |
| (Qi et al., 2023a) | Methodology | Propose a way to evaluate time-to-event accuracy under censoring. | Yes | They propose MAE variants for TTE prediction and explicitly assume independent censoring. |
| (Qi et al., 2024c) | Methodology | Improve calibration of individual survival distributions (D-cal, KM-cal), preserve discrimination, handle censoring correctly in a conformal-regression setting. | Yes | They claim to address discrimination and calibration and evaluate both, censoring assumptions stated and handled. |
| (Qi et al., 2024b) | Methodology | Improve survival probability prediction by post-hoc conformal calibration targeting conditional (covariate-dependent) distribution calibration—aiming to fix both marginal and conditional calibration while preserving discrimination, with theoretical guarantees. | Yes | They claim to address discrimination and calibration and evaluate both thoroughly, censoring assumptions stated and handled. |
| (Qi et al., 2023b) | Methodology | Accurate selection of pertinent features, computation of trustworthy credible intervals (uncertainty), precise estimation of event times or risk scores. | Partially | Metrics match the objective, but censoring bias is not addressed. |
| (Rahman et al., 2023) | Application | Predict risk of death, identify significant prognostic features, compare RSF to Cox and Gradient Boosting on predictive performance (predicting ranking, survivability). | No | Only ranking is evaluated, even though they imply broader prediction goals and censoring not handled. |
| (Rechichi et al., 2024) | Application | Predict disease progression in ALS using EMG-derived metrics, predict survival outcomes (i.e., event occurrence) in ALS, predict survival time (time-to-event), model evolution of ALS over time and generate survival curves to support patient follow-up. | No | Metrics do not match TTE objective, censoring bias not addressed. |
| (Rollo et al., 2024) | Methodology | Distributed synthetic-data survival modeling; claims TTE prediction. | No | Missing TTE metrics, no calibration, and incomplete censoring evaluation. |
| (Seidi et al., 2024) | Application | Determine whether adding geographic location-based public health features improves survival prediction quality, evaluate on SEER data using existing models (CoxPH, DSM), prediction goal is framed strictly in terms of agreement between predicted and observed survival times, quantified by CI (ranking), no claims about predicting time-to-event, calibration, or distribution accuracy. | Yes | They make no distributional or censoring-related claims. |

| Paper | Problem Type | Objectives | Correctly Evaluated? | Explanation |
|---|---|---|---|---|
| (Shen & Chen, 2026) | Methodology | Predict cumulative incidence functions (CIFs) for competing risks at the individual level, make interpretable predictions, provide a flexible neural estimator generalizing Aalen-Johansen. | Yes | Correct metrics for objectives and censoring bias addressed. |
| (Sluiskes et al., 2024) | Application | Predict residual life (survival time) via AFT models, convert residual life into predicted biological age, evaluate biological age clocks using discrimination and calibration metrics. | Yes | Metrics match risk prediction goal, and they correct for random censoring using Uno's CI. |
| (Song et al., 2024) | Application | Improve multimodal cancer survival prediction (prognostication/stratification) from whole-slide histology images and transcriptomics by compressing each modality into a small set of learned prototypes, enabling efficient fusion with lower compute/memory while improving interpretability and predictive performance. | No | They measure only discrimination aspects, and censoring bias not addressed. |
| (Steinberg et al., 2024) | Methodology | Predict the full time-to-event distribution. Improve discrimination, calibration. | Partially | Discrimination and calibration metrics match goals, but they do not evaluate TTE accuracy, despite claiming to predict TTE. |
| (Supriya & Anitha, 2024) | Application | Predict risk ranking for bladder cancer patients, rank patients by risk, provide treatment recommendations using hazard ratios/log risk differences, evaluate predictive performance of DeepMLPSurv vs Cox and other models, analyze tumor recurrence based on features. | Partially | Metrics align with goal, but censoring-robust metrics are missing. |
| (Tan et al., 2023) | Methodology | Evaluate whether GAN-based oversampling improves balanced survival prediction, assess impacts on survival-prediction performance. | No | Missing TTE metrics and calibration, censoring bias is not addressed. |
| (Tang et al., 2026) | Methodology | Enable efficient end-to-end cancer survival prediction from low-resolution whole-slide pathology images by performing zero-shot tumor-microenvironment (TME) segmentation to capture prognostic tissue information without expensive high-resolution patch processing, improving efficiency and interpretability. | No | They measure only discrimination aspects, censoring bias not addressed. |
| (Tang et al., 2023) | Methodology | Improve survival risk ranking using integrative deep model; stratify patients. | Partially | Ranking metrics match objective, but censoring bias not addressed. |
| (Tang et al., 2025) | Methodology | Predict survival probabilities dynamically over time, model longitudinal measurements, improve prediction accuracy over existing longitudinal survival models, interpretability via SHAP. | No | Missing TTE metrics and censoring bias not addressed. |
| (Van Ness et al., 2023) | Application | Predict incident heart failure risk using censored EHR time-to-event data, provide an interpretable survival model using EBMs, deliver accurate survival predictions compared with Cox and other survival models. | Yes | AUC fits the discrimination goal, IBS partially covers calibration, and IPCW applied. |
| (Vauvelle et al., 2023) | Methodology | Improve risk ranking performance over CoxPH, handle censoring properly, enable top-k risk prediction. | Partially | The CI fits the discrimination goal, but censoring bias left unaddressed and their datasets include heavy censoring. |
| (Wan et al., 2024) | Application | Develop time-to-event risk prediction models for melanoma metastatic recurrence, identify patients "at the highest risk of distant recurrence, who are therefore most likely to benefit from adjuvant therapy, focus on the methodological approach to time-to-event prediction. | Partially | Metrics match their TTE and risk goal, but censoring bias not addressed. |
| (Wang et al., 2025b) | Application | Forecast when a financial risk event will occur in a decentralized, privacy-preserving setting, provide explainability, via time-dependent coefficients across forecasting horizons. | No | Missing TTE metrics, censoring bias not addressed. |
| (Wang et al., 2024) | Methodology | Predict time-to-event (specifically time to death); estimate individualized survival curves; predict survival time "as close as possible" to true time; integrate explainability for understanding factors influencing survival time; support clinical time-dependent decision-making. | No | Metrics do not match TTE objective, censoring bias not addressed. |

| Paper | Problem Type | Objectives | Correctly Evaluated? | Explanation |
|---|---|---|---|---|
| (Wang et al., 2025a) | Methodology | Develop a model-averaging method for Cox models with time-varying covariate effects, improve prediction of survival outcomes when the true model structure is unknown or partially misspecified, provide theoretical justification (asymptotic optimality) and demonstrate predictive performance via simulations and real data. | Yes | Metrics align with their goals, and censoring handled stated via IPCW. |
| (Woodward et al., 2024) | Methodology | Predict survival times, predict full survival distributions, handle censoring with novel error and accuracy updates, detect complex genetic associations (epistasis, heterogeneity), provide interpretable rule-based models, perform well across varying censoring proportions, feature dimension, and interaction structure. | No | Missing TTE metrics, missing calibration, missing ranking, incomplete censoring evaluation. |
| (Wu et al., 2024) | Application | Improve cancer survival prediction from whole-slide images by explicitly modeling tumor/tissue heterogeneity as a heterogeneous graph, incorporating pathology domain knowledge about prognostic tissues and their spatial interactions. | No | They evaluate only discrimination, and censoring bias not addressed. |
| (Wu et al., 2023) | Methodology | Using frailty models to incorporate a multiplicative random effect to capture unobserved heterogeneity. | Yes | IBS match their goals, and they correct for censoring using IPCW. |
| (Xie et al., 2026) | Methodology | Improve risk prediction for cancer survival, provide reliable risk predictions via confidence-aware modeling, improve multimodal fusion through cross-modal alignment. | Partially | Metrics align with the goals, but censoring bias not addressed. |
| (Xu et al., 2024) | Application | Predict time-to-event outcomes, specifically time to Alzheimer's Disease onset, show improved time-to-event prediction when combining modalities. | No | Metrics do not match TTE objective, censoring bias not addressed. |
| (Xu & Guo, 2023) | Methodology | Build a nonlinear survival model compatible with Cox's partial likelihood, improve predictive ranking performance compared to CPH, RSF, DeepSurv, provide intrinsic interpretability through NAM shape functions. | Partially | Harrell CI is correct for ranking, but the censoring bias is not addressed. |
| (Yan et al., 2024) | Methodology | Show better survival prediction than baselines, and outperform SOTA based on CI. | Partially | Ranking metrics match objective, but censoring bias not addressed. |
| (Yang & Qiu, 2025) | Methodology | Model the time distributions of the first occurrences of events of interest, estimate the probability mass function (PMF) of the first hitting time. | No | TTE objective is only partially evaluated (MAE only on uncensored cases), censoring is not correctly handled as Antolini CI is biased under random censoring. |
| (Zhang et al., 2025) | Methodology | Predict the full time-to-event distribution via a PDF and survival function S(t), improve prediction of survival time via modeling mean lifetime, accurately predict censoring via novel loss terms, handle static + dynamic (longitudinal) covariates. | Yes | Metrics align with their goals, and they consider MAE-Hinge for censored cases. |
| (Zhang et al., 2024b) | Methodology | Predict time-to-event outcomes using a non-linear Cox extension; model intends to capture non-linear relationships in hazard while also providing interpretable rules; authors state goal of predicting time until an event occurs and emphasize learning hazards, survival function, and membership thresholds that reflect when patients are at high risk; aims to balance predictive performance with interpretability. | No | Metrics do not match TTE objective, censoring bias not addressed. |
| (Zhang et al., 2026) | Methodology | Develop a multimodal framework (MoE + cross-modal attention) to predict survival risk, improve integration of WSI features and pathway-based genomic features, achieve better risk stratification and prognostic discrimination across TCGA cancers. | Partially | They use correct metric for risk ranking, but censoring bias not addressed. |
| (Zhang et al., 2024c) | Application | Predict cancer survival, by integrating of pathological images and genomic data. | No | They claim to do survival prediction, but only use Harrell CI; censoring bias not addressed. |
| (Zhao et al., 2024) | Methodology | Benchmark predictive performance of survival models. | Partially | Metrics match objective, but censoring bias is not addressed. |

| Paper | Problem Type | Objectives | Correctly Evaluated? | Explanation |
|---|---|---|---|---|
| (Zhou et al., 2025) | Methodology | Risk ranking in cancer prognosis; Improve discrimination and generalization of multimodal survival models. | Partially | Metrics match ranking objective, but censoring not properly handled (*e.g.*, no IPCW). |
| (Zisser & Aran, 2024) | Methodology | Predict exact time-to-event (time to deterioration to CKD Stage 5), improve MAE vs. baseline TTE models (explicit claim), predict full survival curves, handle censored data correctly, enable ranking and risk stratification, enable clinical interpretability. | Partially | Metrics match some of their objectives (MAE for TTE accuracy, CI for ranking), but the censoring bias is unaddressed. |

## B. Formal Metric Definitions

Throughout, let $\mathcal{D} = \{(\boldsymbol{x}_i, t_i, \delta_i)\}_{i=1}^N$, where $t_i = \min\{e_i, c_i\}$ and $\delta_i = \mathbb{1}[e_i \leq c_i]$. Let $\widehat{S}(t \mid \boldsymbol{x}_i)$ denote the predicted survival function and let $G(t \mid \boldsymbol{x}) = \Pr(C > t \mid \boldsymbol{X} = \boldsymbol{x})$ denote the conditional censoring survival function. We write $\widehat{G}(t) = \widehat{\Pr}(C > t)$ for a marginal estimate of the censoring survival function, and $\widehat{G}(t \mid \boldsymbol{x})$ for a covariate-conditional estimate. When needed, let $\widehat{r}_i$ denote a scalar predicted risk score, where larger values indicate earlier predicted event.

**Harrell's C-index.** Harrell's C-index (Harrell Jr. et al., 1996) estimates the proportion of concordant pairs among observed comparable pairs:

$$\widehat{C}_{\text{Harrell}} = \frac{\sum_{i,j} \delta_i \, \mathbb{1}[t_i < t_j] \, \mathbb{1}[\widehat{r}_i > \widehat{r}_j]}{\sum_{i,j} \delta_i \, \mathbb{1}[t_i < t_j]}. \tag{1}$$

It evaluates discrimination only.

**Uno's C-index.** Uno's C-index (Uno et al., 2011) uses IPCW to reweight comparable pairs:

$$\widehat{C}_{\text{Uno}}(\tau) = \frac{\sum_{i,j} \delta_i \, \frac{1}{\widehat{G}(t_i)^2} \, \mathbb{1}[t_i < t_j, \, t_i < \tau] \, \mathbb{1}[\widehat{r}_i > \widehat{r}_j]}{\sum_{i,j} \delta_i \, \frac{1}{\widehat{G}(t_i)^2} \, \mathbb{1}[t_i < t_j, \, t_i < \tau]}. \tag{2}$$

With marginal $\widehat{G}(t) = \widehat{\Pr}(C > t)$, this targets random censoring; with conditional $\widehat{G}(t \mid \boldsymbol{x})$, it can target independent censoring if the censoring model is correctly specified.

**Antolini's C-index.** Antolini's C-index (Antolini et al., 2005) compares predicted survival probabilities at the earlier observed event time:

$$\widehat{C}_{\text{Antolini}} = \frac{\sum_{i,j} \delta_i \, \mathbb{1}[t_i < t_j] \, \mathbb{1}\left[\widehat{S}(t_i \mid \boldsymbol{x}_i) < \widehat{S}(t_i \mid \boldsymbol{x}_j)\right]}{\sum_{i,j} \delta_i \, \mathbb{1}[t_i < t_j]}. \tag{3}$$

It evaluates discrimination for predicted survival curves.

**Brier score and IBS.** At a fixed time $t^*$, the IPCW Brier score (Brier, 1950; Graf et al., 1999; Gerds & Schumacher, 2006) is

$$\widehat{\text{BS}}(t^*) = \frac{1}{N} \sum_{i=1}^N \left[ \frac{\widehat{S}(t^* \mid \boldsymbol{x}_i)^2 \mathbb{1}[t_i \leq t^*, \delta_i = 1]}{\widehat{G}(t_i)} + \frac{(1 - \widehat{S}(t^* \mid \boldsymbol{x}_i))^2 \mathbb{1}[t_i > t^*]}{\widehat{G}(t^*)} \right]. \tag{4}$$

The integrated Brier score (Graf et al., 1999) averages this over a time interval:

$$\widehat{\text{IBS}} = \frac{1}{t_{\max}} \int_0^{t_{\max}} \widehat{\text{BS}}(t) \, \mathrm{d}t. \tag{5}$$

**Mean Absolute Error.** Given an ISD, a point event-time prediction can be obtained from the predicted median survival time,

$$\widehat{e}_i = \inf\{t : \widehat{S}(t \mid \boldsymbol{x}_i) \leq 0.5\}. \tag{6}$$

For uncensored observations, MAE is

$$\widehat{\text{MAE}}_{\text{uncens}} = \frac{1}{\sum_i \delta_i} \sum_{i=1}^{N} \delta_i \, |t_i - \widehat{e}_i|. \tag{7}$$

This ignores censored instances. Censored variants, such as MAE-Margin (Haider et al., 2020) and MAE-PO (Qi et al., 2023a), attempt to incorporate censored observations by accounting for the unobserved event times after censoring.

**1-Calibration.** 1-Calibration (D'Agostino & Nam, 2003) evaluates calibration at a fixed horizon $t^*$. Let

$$\widehat{F}(t^* \mid \boldsymbol{x}_i) = 1 - \widehat{S}(t^* \mid \boldsymbol{x}_i)$$

denote the predicted event probability by $t^*$. Instances are sorted by $\widehat{F}(t^* \mid \boldsymbol{x}_i)$ and partitioned into $K$ groups $\mathcal{D}_1, \ldots, \mathcal{D}_K$. A Hosmer–Lemeshow-type statistic is then

$$\chi^2_{\text{HL}} = \sum_{k=1}^{K} \frac{N_k \left( F_k(t^*) - \widehat{F}_k(t^*) \right)^2}{\widehat{F}_k(t^*) \left( 1 - \widehat{F}_k(t^*) \right)}, \tag{8}$$

where $N_k = |\mathcal{D}_k|$, $F_k(t^*)$ is the observed event frequency by $t^*$ in group $k$, and $\widehat{F}_k(t^*)$ is the average predicted event probability in that group.

**D-Calibration.** D-Calibration (Haider et al., 2020) evaluates whether the predicted survival probabilities at event times,

$$\widehat{S}(e_i \mid \boldsymbol{x}_i),$$

are uniformly distributed on $[0, 1]$ under a calibrated model. In practice, the interval $[0, 1]$ is partitioned into bins and the observed bin counts are compared to the expected uniform counts using a goodness-of-fit test. Censored observations are typically distributed over the range of survival probabilities compatible with their censoring time.

**Log-likelihood.** If the model provides both a predicted density $\widehat{f}(t \mid \boldsymbol{x}_i)$ and survival function $\widehat{S}(t \mid \boldsymbol{x}_i)$, the right-censored log-likelihood is

$$\ell = \sum_{i=1}^{N} \left[ \delta_i \log \widehat{f}(t_i \mid \boldsymbol{x}_i) + (1 - \delta_i) \log \widehat{S}(t_i \mid \boldsymbol{x}_i) \right]. \tag{9}$$

It evaluates the full predictive distribution, but requires access to a density or hazard representation.

## C. Proper Scoring Rules

This definition is adapted from (Rindt et al., 2022).

A scoring rule $R$ takes a distribution $P$ over some set $\mathcal{Y}$ and an observed value $y \in \mathcal{Y}$, and returns a score $R(P, y)$ assessing how well the predictive distribution $P$ assigns probability to the observed value. For a positive scoring rule, a higher score indicates a better model fit.

A scoring rule is called *proper* if the true distribution achieves the optimal expected score, *i.e.*, if

$$\mathbb{E}_{y \sim P} R(P, y) \ \geq \ \mathbb{E}_{y \sim P} R(\widehat{P}, y)$$

for all predictive distributions $\widehat{P}$.

In the context of survival analysis, let $S(t \mid \boldsymbol{x})$ denote the true conditional survival function, let $\widehat{S}(t \mid \boldsymbol{x})$ denote a predicted survival function, and let $G(t \mid \boldsymbol{x}) = \Pr(C > t \mid \boldsymbol{X} = \boldsymbol{x})$ denote the conditional censoring survival function. We call the scoring rule $R$ proper if, for every true conditional survival function $S(t \mid \boldsymbol{x})$, every conditional censoring survival function $G(t \mid \boldsymbol{x})$, and every covariate value $\boldsymbol{x} \in \mathcal{X}$, it holds that

$$\mathbb{E}_{E,C \mid \boldsymbol{X}=\boldsymbol{x}} R\big( S(\cdot \mid \boldsymbol{x}), (T, \Delta) \big) \geq \mathbb{E}_{E,C \mid \boldsymbol{X}=\boldsymbol{x}} R\big( \widehat{S}(\cdot \mid \boldsymbol{x}), (T, \Delta) \big)$$

for every predicted survival function $\widehat{S}(\cdot \mid \boldsymbol{x})$, where $E \mid \boldsymbol{X} = \boldsymbol{x}$ has survival function $S(t \mid \boldsymbol{x})$, $C \mid \boldsymbol{X} = \boldsymbol{x}$ has censoring survival function $G(t \mid \boldsymbol{x})$, and $T = \min\{E, C\}$, $\Delta = \mathbb{1}[E \leq C]$.

By taking the expectation with respect to $\boldsymbol{X}$, we obtain

$$\mathbb{E}_{E,C,\boldsymbol{X}} R\big(S(\cdot \mid \boldsymbol{X}), (T, \Delta)\big) \geq \mathbb{E}_{E,C,\boldsymbol{X}} R\big(\widehat{S}(\cdot \mid \boldsymbol{X}), (T, \Delta)\big)$$

for every family of predicted survival functions $\widehat{S}(\cdot \mid \boldsymbol{x})$, with $\boldsymbol{x} \in \mathcal{X}$.

## D. Simulation Details

We evaluate model-metric consistency using a controlled synthetic data-generating process with known ground truth event times. Our simulation framework follows the linear Weibull construction of Foomani et al. (2023), extended to explicitly control the censoring mechanism under random, independent, and dependent censoring mechanisms.

**Covariates.** For each experiment, we generate $N$ independent samples $\{\boldsymbol{x}_i\}_{i=1}^N$, where

$$\boldsymbol{x}_i \sim \text{Unif}([0, 1]^d),$$

with $d = 10$ covariates and $N = 10{,}000$ instances. Covariates are fixed across censoring mechanisms within each random seed (0-99) to isolate the effect of censoring.

**Event and censoring time models.** Both the event time $E$ and censoring time $C$ are generated from Weibull distributions with covariate-dependent hazards. Specifically, conditional on covariates $\boldsymbol{x}_i$, the hazard and survival functions for $E$ (or censoring $C$ respectively) are

$$h_E(t \mid \boldsymbol{x}_i) = \left(\frac{v}{\rho}\right) \left(\frac{t}{\rho}\right)^{v-1} \exp\left(g_\Psi(\boldsymbol{x}_i)\right), \tag{10}$$

$$S_E(t \mid \boldsymbol{x}_i) = \exp\left(-\left(\frac{t}{\rho}\right)^v \exp\left(g_\Psi(\boldsymbol{x}_i)\right)\right), \tag{11}$$

where $v$ and $\rho$ represent the shape and scale parameters of the Weibull distribution, respectively. In all experiments we use

$$(v_E, \rho_E) = (4, 17), \qquad (v_C, \rho_C) = (3, 12).$$

The risk functions $g_E(\boldsymbol{x})$ and $g_C(\boldsymbol{x})$ are linear:

$$g_E(\boldsymbol{x}) = \boldsymbol{x}^\top \boldsymbol{\beta}_E, \qquad g_C(\boldsymbol{x}) = \boldsymbol{x}^\top \boldsymbol{\beta}_C,$$

with coefficients $\boldsymbol{\beta}_E, \boldsymbol{\beta}_C \in \mathbb{R}^d$ sampled independently for each random seed from $\text{Unif}([-1, 1]^d)$.

**Copula-based dependence.** To control dependence between $E$ and $C$, we adopt a copula-based construction. For each instance $i$, we first sample a pair of uniform random variables $(U_i, V_i) \in [0, 1]^2$ from an Archimedean Clayton copula with Kendall's $\tau \in \{0.25, 0.5, 0.75\}$. The copula parameter $\theta$ is chosen to match the desired Kendall's $\tau$ via the standard Clayton relationship. Event and censoring times are then obtained via inverse transform sampling:

$$E = S_E^{-1}(V_i \mid \boldsymbol{x}_i), \qquad C = S_C^{-1}(U_i \mid \boldsymbol{x}_i).$$

When $\tau = 0$, $U_i$ and $V_i$ are sampled independently, yielding marginal independence between $E$ and $C$.

**Censoring mechanisms.** We consider three censoring mechanisms:

- **Random censoring.** Censoring times are generated independently of both $\boldsymbol{X}$ and $E$. Formally, $E \perp\!\!\!\perp C$.

- **Independent censoring.** Censoring depends on covariates but remains conditionally independent of the event time: $E \perp\!\!\!\perp C \mid \boldsymbol{X}$.
  This is achieved by letting $g_C(\boldsymbol{x})$ depend on $\boldsymbol{x}$ while sampling $(U, V)$ independently.

- **Dependent censoring.** Censoring and event times are dependent even after conditioning on covariates: $E \not\!\perp\!\!\!\perp C \mid \boldsymbol{X}$.
  This dependence is induced via the Clayton copula with $\tau > 0$.

**Observed data.** The observed time and event indicator are given by

$$t_i = \min(e_i, c_i), \qquad \delta_i = \mathbb{I}\{e_i \le c_i\}.$$

For oracle evaluation, we retain access to the true event times $e_i$ for all instances.

**Model fitting and evaluation.** For each dataset, we fit a CoxPH model using the observed $(t_i, \delta_i, \boldsymbol{x}_i)$ and obtain predicted survival curves $\widehat{S}(t \mid \boldsymbol{x}_i)$ via the Breslow estimator. Model performance is evaluated in three ways: (1) *oracle* metrics computed from true event times, (2) *naive* metrics computed directly from the uncensored instances without correction, and (3) *IPCW-corrected* metrics using estimated censoring weights from the KM estimator. All metrics are computed on held-out test sets and averaged over 100 random seeds. We additionally report the empirical censoring rate and the number of events to quantify censoring severity.

