# OpenReview forum: "Position: Stop Chasing the C-index when Evaluating Survival Analysis Models"
_ICML.cc/2026/Position_Paper_Track — ICML 2026 Position Paper Track spotlight_

### Official Review · Reviewer_nTaE · 2026-02-23

**Significance:** 3
**Argument Clarity:** 3
**Rating:** 5
**Confidence:** 3

**Questions:**

How did you sample the 92 papers (venues, inclusion/exclusion, time window)? Any reason the sample might over-represent certain subfields?

**Alternative Views Section:**

Yes

**Compliance With Llm Reviewing Policy A Conservative:**

Affirmed.

**Discussion Potential:**

3

**Final Justification:**

The authors addressed my concern, I increased my score to accept.

**Paper Summary:**

This position paper argues that evaluation practice in survival analysis is often mis-specified: researchers frequently default to discrimination metrics (especially the C-index) even when the stated scientific goal is time-to-event accuracy or probabilistic calibration, and they often leave censoring assumptions implicit or unjustified. The authors support this claim with (i) a survey of 92 survival-analysis papers (2023–2025) reporting substantial objective–metric misalignment, and (ii) an empirical demonstration that commonly used estimators for metrics such as the C-index and integrated Brier score can become biased under violations of (conditional) independent censoring. They propose desiderata for selecting metrics, introduce a “double-helix ladder” framing that emphasizes model–metric–assumption alignment, discuss alternative viewpoints, and end with practical recommendations and a call for better evaluation methodology under dependent censoring.

**Position:**

Yes

**Position In Title:**

Yes

**Related Work:**

3

**Strengths And Weaknesses:**

Pros:
- Clear position. The author's central stance (“stop chasing the C-index” as a default) is explicit and consistently reinforced throughout the paper. Other aspects of the track are fulfilled (e.g., alternative views)
- Support via reasoning + evidence. The argument is grounded in censoring-specific identifiability and bias considerations, and it is backed by a literature survey and controlled experiments illustrating bias when censoring assumptions are violated.
- The paper is well written and easy to follow


Cons:
- The “double-helix ladder” would benefit from a more concrete decision aid. As presented, it sounds conceptually appealing, but readers will likely want a crisp mapping to recommended metrics/estimators, with explicit “valid / invalid / requires sensitivity analysis” guidance.
- Related work could be expanded, specifically, methodological work under dependent censoring (e.g., sensitivity analysis/ partial identification approaches as in https://arxiv.org/abs/2510.13397)
- Some aspects regarding the evidence/ literature sample remain unclear (see questions)

**Support:**

3

---

> ### Author Rebuttal · Authors · 2026-03-30
>
> We thank the reviewer for reading our paper and giving us their suggestions. Below we reply to each comment individually.
>
> > The “double-helix ladder” would benefit from a more concrete decision aid. As presented, it sounds conceptually appealing, but readers will likely want a crisp mapping to recommended metrics/estimators, with explicit “valid / invalid / requires sensitivity analysis” guidance.
>
> Thank you for this comment. We agree that the current presentation may make the recommendations harder to follow in practice. To improve clarity, we will introduce a decision-tree style summary that consolidates the guidance into a single structured framework, with explicit mapping from setting to recommended metrics and estimators. This will be placed immediately after the practical recommendations section. We refer to our response to Reviewer C41S for a preview.
>
> > Related work could be expanded, specifically, methodological work under dependent censoring (eg, sensitivity analysis or partial identification approaches)
>
> Thank you for this suggestion. We agree that methodological work under dependent censoring is an important area, and we will expand the related work (Section 5.  Dependent Censoring) in the revised manuscript. Existing approaches to dependent censoring fall into three categories: (1) sensitivity analyses, which assess how conclusions vary under different assumptions about the censoring mechanism; (2) partial identification methods, which derive bounds under weaker, nonparametric assumptions; and (3) model-based corrections, which explicitly model the joint distribution of event and censoring times.
>
> This section will also discuss a recent approach by [1] replaces the Kaplan-Meier estimator within standard metrics by the Copula-Graphic estimator that jointly models event and censoring times. While empirically promising, such approaches require specification of a copula family – introducing a strong and typically unverifiable assumption – and are currently supported primarily by simulation evidence, with limited theoretical guarantees (eg, consistency or asymptotic normality).
>
> [1] Lillelund et al. Overcoming dependent censoring in the evaluation of survival models. 2025, preprint arXiv:2502.19460.
>
> > [Q1] How did you sample the 92 papers (venues, inclusion/exclusion, time window)? Any reason the sample might over-represent certain subfields?
>
>
> Thank you for this question. We sampled 92 papers published between 2023.01.01–2025.12.31 using keyword-based searches across major databases (Google Scholar, Scopus, ScienceDirect, Springer, IEEE Xplore), combined with manual screening of major ML venues (eg, NeurIPS, ICML, ICLR, AISTATS), selected biomedical conferences (eg, EMBC), and relevant journals. Papers were included if they (i) addressed survival analysis with a learned model and (ii) reported quantitative evaluation metrics; purely methodological statistics papers and epidemiological studies without predictive modeling were excluded.
>
>
> Our strategy intentionally focuses on machine learning-oriented survival analysis. As a result, the sample may over-represent ML and biomedical venues relative to traditional biostatistics, but this aligns with our goal of assessing evaluation practices in modern predictive modeling settings. In the revised manuscript, we will mention this aspect.

---

> > ### Author Rebuttal · Reviewer_nTaE · 2026-04-02
> >
> > Thank you for addressing my points. Given that these will be incorporated into the camera-ready version, I recommend acceptance and adjust my score accordingly.

---

### Official Review · Reviewer_xNBM · 2026-03-08

**Significance:** 3
**Argument Clarity:** 3
**Rating:** 5
**Confidence:** 4

**Questions:**

1. Is there a role for semi-synthetic experiments for addressing censoring, like how the causal inference literature handles hidden confounders?

2. What is a potential avenue for developing better metrics for more advanced forms of censoring?

**Alternative Views Section:**

Yes

**Compliance With Llm Reviewing Policy A Conservative:**

Affirmed.

**Discussion Potential:**

2

**Final Justification:**

The clarifications given by the authors for robust evaluation have reinforced my positive assessment.

**Paper Summary:**

Survival models are overwhelmingly evaluated on concordance-based measures on whether individuals are correctly ranked by risk. These metrics do not always align with prediction accuracy, and more importantly, they do not account for biases from censoring. Methodological developments allow models to account for conditionally independent, rather than purely random, censoring. There is a need to develop consistent evaluations for models under these different censoring assumptions.

**Position:**

Yes

**Position In Title:**

Yes

**Related Work:**

3

**Strengths And Weaknesses:**

This position paper illustrates the stated problem with concrete examples from the literature. These support the argument that the community should pay closer attention to the open challenge of evaluating models with real-world data under potentially arbitrary censoring. The desiderata for appropriate evaluation metrics are sound.

It would be more satisfying if the authors could present a path forward for evaluation under conditionally independent censoring. IPCW correction has the weakness of relying on another model to generate the weights. Engaging with this fundamental limitation would have made the discussion more exciting. The authors have straightforward suggestions for evaluation under random censoring.

More broadly, the authors appear to be making two distinct points: the potential misalignment between ranking performance and predictive accuracy, and the issues with censoring. The practical recommendations and alternative views are interleaved between these two separate points. This hurts the clarity of the paper.

The value of Figure 4 is questionable because the results from the two C-indices appear to be similar.

The double-helix ladder might be a confusing way to present the idea that evaluation metrics have lagged behind survival models in dealing with censoring.

**Support:**

4

---

> ### Author Rebuttal · Authors · 2026-03-30
>
> We thank the reviewer for carefully reading our paper and providing valuable feedback. To address the concerns.
>
> > It would be more satisfying if the authors could present a path forward for evaluation under conditionally independent censoring.
>
> We agree that IPCW relies on an additional model, as noted in lines 318-323. Consequently, IPCW inherits the limitations of that model. For example, the KM does not account for covariates, and Cox-based weighting is only asymptotically consistent when the proportional hazards assumption holds. More broadly, misspecification of the censoring model can directly affect the resulting evaluation.
>
> That said, IPCW remains, to our knowledge, one of the most theoretically principled approaches currently available for evaluation under independent censoring. While it is imperfect, there is still no broadly accepted alternative that is clearly better in the general case of conditionally independent censoring.
>
> We agree that the original manuscript did not provide a concrete recommendation on this issue. In the revision, we will clarify that IPCW-based evaluation should be accompanied by sensitivity analysis. Specifically, one can estimate IPCW weights using multiple censoring models (ideally, with different underlying assumptions) and then examine whether the resulting metric values, especially model rankings, are consistent across these choices. Stability across weighting models would strengthen confidence in the conclusions, whereas large differences would indicate that the evaluation is sensitive to assumptions about the censoring process and should be interpreted more cautiously. We will add this point to the manuscript as a practical path forward for evaluation under conditionally independent censoring.
>
>
> > The practical recommendations and alternative views are interleaved between these two separate points. This hurts the clarity.
>
> We agree and will introduce a decision-tree style summary that consolidates the practical guidance into a single structured framework. This will be placed immediately after the practical recommendations section. Due to the space limit, please refer to our [W1] response to Reviewer C41S.
>
> > Clarity of double helix ladder.
>
> Thank you for pointing this out. We will revise the figure by adding side annotations to the blobs that highlight representative models and metrics under each censoring assumption. We will also include the year associated with each method or metric to support the illustration of “lagging”. For example, both the CoxPH and IPCW-IBS assume independent censoring, but CoxPH was introduced in the 1970s, whereas the IPCW-IBS metric was proposed in 1999.
>
> > [Q1] Is there a role for semi-synthetic experiments for addressing censoring?
>
> Thank you for this question. Yes, semi-synthetic experiments are useful because real data does not reveal true event times for censored instances. They allow controlled variation of censoring mechanisms while retaining realistic structure from real data.
>
> However, their usefulness depends on realism.. They should be anchored to real covariates and event distributions whenever possible. For example, Qi et al. (2023) learns survival and censoring times from real data, and impose synthetic censoring mechanisms to enable controlled evaluation of MAE-type metrics. Foomani et al. (2023) creates semi-synthetic datasets with dependent censoring from copulas.
>
> We therefore see semi-synthetic experiments as a useful complement to real-data studies: **not a substitute, but a principled way to perform sensitivity analyses under realistic alternative censoring mechanisms and to show improvement of censoring-aware models in a controlled setting**.
>
> Qi et al. An effective meaningful way to evaluate survival models. ICML, 2023.
>
> Foomani et al. Copula-Based Deep Survival Models for Dependent Censoring. UAI, 2023.
>
>
> > [Q2] What is a potential avenue for developing better metrics for more advanced forms of censoring?
>
> Thank you for this question. A promising direction is to develop estimators that account for dependence between event and censoring times, ranging from simple dependence parameterizations to nonlinear approaches such as copulas. At the same time, metrics should avoid strong parametric assumptions on the censoring model. For example, relying on a CoxPH model for the censoring process implicitly assumes proportional hazards, which may not hold in practice. We therefore see value in approaches that either (1) use flexible, potentially nonparametric estimators of the censoring mechanism, or (2) explicitly study robustness to misspecification of that mechanism.
>
> Because the true event times for censored instances are unobserved in real data, developing such metrics requires careful validation. Semi-synthetic data provide a natural framework, enabling controlled variation of the censoring mechanism while preserving realistic covariates and event distributions.

---

> > ### Author Rebuttal · Reviewer_xNBM · 2026-04-01
> >
> > Thank you for the thoughtful response. The arguments presented for sensitivity analysis and semi-synthetic evaluation are acknowledged. I have one follow-up question regarding sensitivity analyses.
> >
> > > In the revision, we will clarify that IPCW-based evaluation should be accompanied by sensitivity analysis. Specifically, one can estimate IPCW weights using multiple censoring models (ideally, with different underlying assumptions) and then examine whether the resulting metric values, especially model rankings, are consistent across these choices. Stability across weighting models would strengthen confidence in the conclusions, whereas large differences would indicate that the evaluation is sensitive to assumptions about the censoring process and should be interpreted more cautiously. We will add this point to the manuscript as a practical path forward for evaluation under conditionally independent censoring.
> >
> > Can you clarify how this would work in practice? It seems unfair to compare the same set of survival models across different censoring assumptions, as different models are useful for different kinds of censoring. On the other hand, if you "adjust" the survival model according to the same censoring models that are used for evaluation, that is not fair either.

---

### Official Review · Reviewer_kDpi · 2026-03-14

**Significance:** 2
**Argument Clarity:** 2
**Rating:** 4
**Confidence:** 3

**Questions:**

Please see the weakness section and respond to the concerns raised in the weaknesses.

**Alternative Views Section:**

Yes

**Compliance With Llm Reviewing Policy A Conservative:**

Affirmed.

**Discussion Potential:**

3

**Final Justification:**

The responses from the reviewers around my core concerns of identifiability are satisfactory.

**Paper Summary:**

The paper has argued that most of the survival analysis literature is heavily reliant on comparisons that often suffer from a mismatch in the modeling assumption and the assumptions behind the metric used. Or sometimes a mismatch between the metric used and the targeted goals of the study. Concordance index (C-index) is a very commonly used measure. The authors survey 92 papers and found that most of them try to support claims on time-to-event prediction or calibration that C-index does not assess. They introduce five desiderata for a metric namely: proper scoring rule, interpretability, model agnosticism, sensitivity to miscalibration, robustness to censoring. They propose a "Ladder Hypothesis of Metric Consistency" requiring that the models and metrics operate under the same assumptions.  Through controlled experiments, the authors demonstrate the value of the proposed ladder. The paper also gives practical recommendations that state researchers should select metrics aligned with research objectives.

**Position:**

Yes

**Position In Title:**

Yes

**Related Work:**

3

**Strengths And Weaknesses:**

## Strengths

The submission analyzes an important issue with many survival analysis papers. The authors conduct a survey of 92 papers on survival analysis and conclude that many of the papers use C-index as a metric. They conclude that 73 percent of the papers use metrics not aligned with their research objectives.  The alternative views section provides two interesting views and provides credible defence for both. Overall, the paper is clearly written and easy to follow.

## Weaknesses

  1. **Issues around Identifiability**
The paper's central premise, which is that we must align metric assumptions with model assumptions, ignores a deeper gap.
While the authors focus on the "metric-model gap," they gloss over the "model-data gap." Following Tsiatis (1975), we cannot conclude if the data follows a dependent censoring or independent censoring distribution. Since we can never verify if reality is governed by independent or dependent censoring, any placement on the rung is a leap of faith. The promise of "trustworthy evaluation" is built on this untestable assumption. Suppose model A follows independent censoring assumption, metric X follows independent censoring assumption, model B follows dependent censoring assumption and true data follows dependent censoring assumption. In this case, model A and metric X are on the same rung but inconsistent with data. Model B and metric X are not on the same rung but model B is better aligned with true data. This creates a fundamental mismatch. Based on authors suggestion one should use Model A and metric X but that is clearly wrong.


  2. **Lack of real-data demonstration**  The paper critiques many real papers. It would have been more insightful if reviewers showed how model rankings flipped under different metrics more aligned with actual objectives of the research.

3.  **Desiderata are not well justified**  D2 (interpretability) is not well justified and subjective.

**Support:**

2

---

> ### Author Rebuttal · Authors · 2026-03-30
>
> We thank the reviewer for these comments and suggestions. To address the concerns:
>
> > Issues around Identifiability. The paper's central premise, which is that we must align metric assumptions with model assumptions, ignores a deeper gap. While the authors focus on the "metric-model gap," they gloss over the "model-data gap." Following Tsiatis (1975), we cannot conclude if the data follows a dependent censoring or independent censoring distribution.
>
> Thank you for this thoughtful critique. We agree that, ideally, assumptions about the data, model, and metric should all be aligned. However, our claim is more limited: evaluation is only meaningful when the assumptions of the model and the metric are aligned. This is distinct from identifying the true data-generating process, which, as noted by Tsiatis (1975), is not possible in general.
>
> Regarding the reviewer’s example:
>
> Model A + Metric X are aligned but inconsistent with the true data-generating process, while Model B is consistent with the true data but misaligned with the metric. If Model B is evaluated using a metric that assumes random/independent censoring, then the evaluation itself is invalid (eg, biased or inconsistent). Therefore, one cannot conclude from that metric that Model B is better. The issue is not that Model B is inferior in its true performance (see also Table 2 in the paper), but the evaluation procedure is not appropriate for it.
>
> Our framework does not claim to identify which model is closest to the true data-generating process. Rather, it ensures that any comparison between models is logically coherent under the stated assumptions. Without alignment between model and metric, the evaluation itself becomes unreliable (see Figure 4), regardless of how well a model may reflect the underlying (unidentifiable) data-generating process.
>
> > Lack of real-data demonstration. The paper critiques many real papers. It would have been more insightful if reviewers showed how model rankings flipped under different metrics more aligned with actual objectives of the research.
>
> Thank you for this suggestion. Our goal was to isolate and quantify the effects of model-metric mismatch in a controlled setting, where (1) the ground truth is known and (2) the data-generating process is fully transparent and reproducible. This allows us to clearly demonstrate failure modes that are otherwise difficult to verify in real data.
>
> We agree that demonstrating ranking changes on real-world datasets would be valuable. That said, this phenomenon has already been empirically observed – for example, Qi (2023) showed through controlled experiments that model rankings can change when evaluated under different MAE-type metrics and censoring assumptions. Our contribution is complementary: rather than re-demonstrating this effect, we provide a principled explanation of why such ranking changes occur, by linking them directly to misalignment between model and metric assumptions, and validating this mechanism in a setting where the ground truth is known.
>
> Qi et al. An effective meaningful way to evaluate survival models. ICML, 2023.
>
> > Desiderata are not well justified D2 (interpretability) is not well justified and subjective.
>
> Thank you for this suggestion. We agree that interpretability is inherently subjective and that our framing desiderata 2 is not specific enough. Therefore, we have revised D2 in the revised manuscript to make it more concrete and specific to survival analysis. It now reads:
>
> *D2: Interpretability: A metric should be expressed in domain-relevant, human-understandable units (eg, days, weeks, months) and described using common, non-technical terminology. Its value should have a direct and intuitive connection to model behavior, without requiring additional transformations or assumptions. This allows practitioners and non-technical stakeholders to readily understand what is being measured and how differences in the metric reflect differences in model performance.*

---

> > ### Author Rebuttal · Reviewer_kDpi · 2026-04-01
> >
> > The responses from the reviewers around my core concerns of identifiability are satisfactory.

---

### Official Review · Reviewer_C41S · 2026-03-14

**Significance:** 4
**Argument Clarity:** 4
**Rating:** 6
**Confidence:** 5

**Questions:**

1. In Recommendation 1, you bring up the example of prioritizing recipients for organ allocation as being a good fit for evaluation using a C-index. I agree somewhat with the premise--if our objective is indeed to come up with the most accurate ranking of recipients, I don't see how evaluation using a C-index achieves that goal. Since we don't get to observe a single organ being transplanted into different recipients, a C-index would only be ranking outcomes from different organs being transplanted into different recipients, for which the outcomes may depend heavily on the health of the donor organ, which is not comparable across all donors. Furthermore, there have been studies from the transplantation community (see Wolfe et al. (2009) for an example) that are more critical of C-indices as useful measures for evaluation. Given these concerns, I would appreciate it if you could provide more information on why you believe C-indices are useful for organ allocation.
2. If we have no reliable way to evaluate under dependent censoring, are you advocating to stop developing models for the dependent censoring setting until we can catch up with evaluation metrics?

Reference:
- Wolfe, R. A., McCullough, K. P., & Leichtman, A. B. (2009). Predictability of survival models for waiting list and transplant patients: Calculating LYFT. American Journal of Transplantation, 9, 1523-1527. https://doi.org/10.1111/j.1600-6143.2009.02708.x

**Alternative Views Section:**

Yes

**Compliance With Llm Reviewing Policy A Conservative:**

Affirmed.

**Discussion Potential:**

3

**Final Justification:**

I strongly recommend acceptance. The authors have addressed weakness 1 during the rebuttal by proposing recommendations in the form of a decision tree, which seems reasonable to me and potentially helpful for readers.

**Paper Summary:**

The authors consider a very important problem in the area of survival analysis, the usage of the concordance index (C-index) for evaluating survival prediction models. Survival analysis has gained significant attention from the ML community over the last 10 years or so, and from interacting with other researchers in this community, I find that there is near-universal agreement that the C-index is not a good evaluation metric for most cases. Yet, we all still keep using it in our papers, perhaps out of convenience or a desire for comparison against prior work (both of which are discussed as alternative views in this paper).

The authors provide a convincing narrative on why we should stop "chasing" the C-index, i.e., trying to build models that yield a higher C-index, and rather focus on better alignment between evaluation and model development. The paper also provides some practical recommendations that would likely be beneficial to the ML community if adopted.

**Position:**

Yes

**Position In Title:**

Yes

**Related Work:**

4

**Strengths And Weaknesses:**

## Strengths
- Clear identification of a current problem in evaluation of survival prediction models, supported with strong evidence from a survey of recent literature, as well as their own experiments.
- Extremely well written. I would gladly pass this paper to any other researcher working in survival analysis as a reference. This paper seems about halfway to a practical guidebook of publication requirements for survival prediction models, which may be the ultimate goal someday.
- Has the potential to be highly impactful, moving the discussion on evaluation of survival prediction models beyond informal conversations between researchers to a problem worthy of more serious consideration. I think this paper would be highly cited by future papers on survival analysis, particularly to provide justification of the use of alternative metrics beyond C-indices.

## Weaknesses
- Practical recommendations may be too vague and difficult for researchers to implement on their own. A suggested set of recommendations in the form of a decision tree (e.g., if you are using models A, B, or C for tasks D, E, or F, then evaluate using metric M).
- No real recommendation on how to close the gap between model development and evaluation. See my question 2 below.

Minor issues:
- Reference Emura & Chen (2018) is duplicated.
- Line 252: Missing word "is": D-Calibration *is* model-agnostic
- Table 1 is not referred to at any point in the body text.
- I suggest removing the # sign for interpretability in Table 1, as it suggests that the numbers are a ranking, where 1 is best rather than worst.

**Support:**

4

---

> ### Author Rebuttal · Authors · 2026-03-30
>
> Thank you for your review and helpful comments. We have corrected the minor issues you raised. Let us first address the first weakness you have identified and then address your questions.
>
> > W1: Practical recommendations may be too vague and difficult for researchers to implement on their own. A suggested set of recommendations in the form of a decision tree.
>
> Thank you for this suggestion. We agree and will include a decision tree in the revised version of the manuscript. This will appear following the practical recommendations section and follow this flow:
>
> Step 1: What task do I want to solve?
>
> * Ranking -> discrimination metrics.
>
> * Time-to-event prediction -> error-based metrics.
>
> * Survival probability estimation -> calibration metrics.
>
> Step 2: What assumptions will I make about censoring?
>
> * Random (marginal) censoring -> IPCW with marginal estimators (eg, Kaplan-Meier weights)
>
> * Independent censoring (conditional on covariates) -> IPCW with conditional estimators for censoring (covariate-adjusted weights using, for example, CoxPH). Ideally, a sensitivity analysis should be performed to ensure that the conclusion is not susceptible to the limitation of conditional estimators.
>
> * Dependent censoring -> sensitivity analysis (eg, subgroup evaluation with proxy covariates) or assumption-based methods (eg, copula models). **Note: Under dependent censoring, evaluation lacks generally valid metrics.**
>
> Step 3: What metric to use? (given Step 1 + Step 2)
>
> * Under Ranking in Step 1:
>
>  -> Use discrimination metrics (eg, C-index, time-dependent AUC)
>
> -> Apply the censoring adjustment from Step 2 (marginal IPCW / conditional IPCW / sensitivity or adjusted methods)
>
> * Under time-to-event prediction in Step 1:
>
> -> Use error-based metrics (eg, MAE, MSE, IBS)
>
> -> Apply the censoring adjustment from Step 2
>
> * Under survival probability estimation in Step 1:
>
>  -> Use calibration metrics (eg, calibration curves, D-calibration, Brier score)
>
>  -> Apply the censoring adjustment from Step 2.
>
> > [Q1]: Given these concerns, I would appreciate it if you could provide more information on why you believe C-indices are useful for organ allocation.
>
> Thank you for this question. We agree that there are many subtle issues in organ transplantation, as this involves many factors beyond recipient health, including transplant **benefit** (how many more years the recipient will live, versus no transplant), as well as distribution of compatible [donor, recipient] pairs; see LYFT [1].
>
> We will revise the manuscript to frame this more generally as **critical resource allocation**, rather than focusing on organ transplantation specifically. Examples include prioritizing ALS (amyotrophic lateral sclerosis) patients for assisted breathing devices based on predicted respiratory decline, or ranking elderly by fall risk to allocate preventive physiotherapy. In these settings, the C-index evaluates whether higher-risk individuals tend to experience earlier events, which serves as a proxy for prioritization under observed data, but does not directly assess counterfactual benefit. It is therefore most appropriate when differences in outcomes primarily reflect underlying risk and individuals are reasonably comparable.
>
> Lastly, an important note is that C-index weighs all pairwise comparisons equally. As a result, it treats a comparison between candidates with the 1st and 500th longest lifespans the same as a comparison between the 499th and 500th, even though only the former is relevant for allocation decisions. A log C-index would address this limitation by placing greater emphasis on differences that matter most for prioritization. We will mention this in the revised manuscript.
>
> [1] Wolfe et al. Predictability of survival models for waiting list and transplant patients: Calculating LYFT. American Journal of Transplantation, 9, 1523-1527. 2009.
>
> > [Q2]: If we have no reliable way to evaluate under dependent censoring, are you advocating to stop developing models for the dependent censoring setting until we can catch up with evaluation metrics?
>
>
> Thank you for this question. We are not advocating to stop developing models for dependent censoring. Rather, model development should be accompanied by evaluation in settings where ground truth is known. In practice, this includes heterogeneous and realistic synthetic or semi-synthetic data (eg, Weibull or exponential distributions with controlled dependence via copulas), where the true data-generating process is known. If improvements can be established in such controlled settings, this provides evidence that the model gains are not artifacts of the evaluation itself.
>
>
> Our concern is instead with claims of superiority under dependent censoring that are supported by metrics that are only consistent under stronger assumptions. In such cases, it is hard to determine what are actual model improvements and what are artifacts introduced by the evaluation metric.

---

> > ### Author Rebuttal · Reviewer_C41S · 2026-04-01
> >
> > The authors have addressed all of the weaknesses and answered my questions. The proposed decision tree seems reasonable to me and potentially helpful for readers. I continue to support the paper strongly for acceptance.

---

### Decision · Program_Chairs · 2026-04-30

**Decision:**

Accept (spotlight)

**Comment:**

The position is an important point.  I would have thought it would be already accepted, but the paper shows clearly this is not the case and clearly shows the problem.  It is well-written, important, and the reviewers are strongly convinced.  There was great discussion between authors and reviewers that made the reviewers even more positive.